# Ice formation on lake surface in winter causes warm season bias of lacustrine brGDGT temperature estimates

Jiantao Cao [1,2], Zhiguo Rao [3,*], Fuxi Shi [4], Guodong Jia [1,*]

[1] State Key Laboratory of Marine Geology, Tongji University, Shanghai 200092, China

[2] Key Laboratory of Western China's Environmental Systems, Ministry of Education, College of Earth and Environmental Sciences, Lanzhou University, Lanzhou, 730000, China

[3] College of Resources and Environmental Sciences, Hunan Normal University, Changsha, 410081, China

[4] Jiangxi Provincial Key Laboratory of Silviculture, College of Forestry, Jiangxi Agricultural University, Nanchang, 330045, China

[*]*Corresponding authors*: Zhiguo Rao (raozhg@hunnu.edu.cn); Guodong Jia (jiagd@tongji.edu.cn).

**Abstract**

It has been frequently found that lacustrine brGDGT-derived temperatures are warm season biased relative to measured mean annual air temperature (AT) in the mid to high latitudes, the mechanism of which, however, is not very clear. Here, we investigated the brGDGTs from catchment soils, suspended particulate matter (SPM) and surface sediments in different water depths in the Gonghai Lake in north China to explore this question. Our results showed that the brGDGT distribution in sediments resembled that in the SPM but differed from the surrounding soils, suggesting a substantial aquatic origin of the brGDGTs in the lake. Moreover, the increase of brGDGT content and decrease of methylation index with water depth in sediments suggested more contribution of aquatic brGDGTs produced from deep/bottom waters. Therefore, established lake-specific calibrations were applied to estimate local mean annual AT. As usual, the estimates were significantly higher than the measured mean annual AT. However, they were similar to, and thus actually reflected, the mean annual lake water temperature (LWT). Interestingly, the mean annual LWT is close to the measured mean warm season AT, hence suggesting that the apparent warm season bias of lacustrine brGDGT-derived temperatures could be caused by the discrepancy between AT and LWT. In our study region, ice forms at the lake surface during winter, leading to isolation of the underlying lake water from air and hence higher LWT than AT, while LWT basically follows AT during warm seasons when ice disappears. Therefore, we think what lacustrine brGDGTs actually reflected is the mean annual LWT, which is

higher than the mean annual AT in our study location. Since the decoupling between LWT and AT in winter due to ice formation is a universal physical phenomenon in the mid to high latitudes, we propose this phenomenon could be also the reason for the widely observed warm season bias of brGDGT-derived temperatures in other seasonally surface ice-forming lakes, especially the shallow lakes.

**Keywords**: lake sediments, aquatic brGDGTs, temperature proxy, seasonality, ice formation

## 1    Introduction

The branched glycerol dialkyl glycerol tetraethers (brGDGTs), including 0–2 cyclopentyl moieties (a–c) and four to six methyl groups (I–III) (Weijers et al., 2007a), are components of the cell membranes of microorganisms ubiquitously found in marine and continental environments and sensitive to ambient environmental conditions (Sinninghe Damsté et al., 2000; Weijers et al., 2006a; Schouten et al., 2013). The relative amounts of methyl groups and cyclopentyl moieties, expressed as methylation index and cyclization ratio of brGDGTs (such as MBT/CBT or MBT'/CBT) in soil brGDGTs, has been proposed to reflect mean annual air temperature (AT) (Weijers et al., 2007a; Peterse et al., 2012). With improved analytical methods, a series of 6-methyl brGDGTs, previously co-eluted with 5-methyl brGDGTs, were identified (De Jonge et al., 2013), which may introduce scatter in the original MBT'/CBT calibration for the mean annual AT (De Jonge et al., 2014). Thus,

exclusion of the 6-methyl brGDGTs from the MBT', i.e. the newly defined MBT'$_{5ME}$, results in
improved calibrations (De Jonge et al., 2014; Wang et al., 2016; Wang et al., 2019). Calibrations using
globally distributed surface soils for the MBT/CBT, MBT'/CBT or MBT'$_{5ME}$ indices (Weijers et al.,
2007a; Peterse et al., 2012; De Jonge et al., 2014) have been widely used for continental AT
reconstruction (e.g., Weijers et al., 2007b; Niemann et al., 2012; Lu et al., 2019).

BrGDGTs in lake environments were initially thought to be derived from soil input (Hopmans et

al., 2004; Blaga et al., 2009), allowing the mean annual AT to be reconstructed from lake sediments.
However, when the soil-based calibrations are applied to the lake materials, the estimated
temperatures are usually significantly lower than actual local AT (Tierney and Russell, 2009; Tierney
et al., 2010; Blaga et al., 2010; Loomis et al., 2011, 2012; Pearson et al., 2011; Sun et al., 2011;
Russell et al., 2018), suggesting an intricate brGDGTs response to ambient temperature in aquatic
environments. Later, more and more studies reveal that brGDGTs could be produced in situ in lake
environments and differ significantly from soil derived brGDGT distributions (Wang et al., 2012;
Loomis et al., 2014; Naeher et al., 2014; Hu et al., 2015; Cao et al., 2017) and stable carbon isotope
composition (Weber et al., 2015, 2018). The findings of intact polar lipid of brGDGTs, indicative of
fresh microbial products, in lake water suspended particulate matter (SPM) and surface sediments
(Tierney et al., 2012; Schoon et al., 2013; Buckles et al., 2014a; Qian et al., 2019) further confirm the
in-situ production of brGDGTs. Nevertheless, the brGDGT distribution in lake surface sediments has
been found to be still strongly correlated with AT. Subsequently, quantitative lacustrine-specific
calibrations for AT have been established at regional and global scales (Tierney et al., 2010; Pearson
et al., 2011; Sun et al., 2011; Loomis et al., 2012; Shanahan et al., 2013; Foster et al., 2016; Dang et
al., 2018; Russell et al., 2018), which have been widely used for AT reconstruction. These
lacustrine-specific calibrations may reflect mean annual AT well in low-latitude regions (Tierney et al.,
2010; Loomis et al., 2012), such as in the Lake Huguangyan (21°09′ N, 110°17′ E) in south China (Hu
et al., 2015), Lake Donghu (30°54′ N, 114°41′ E) in central China (Qian et al., 2019) and Lake Towuli
(2.5° S, 121° E) on the island of Sulawesi (Tierney and Russell, 2009). However, they usually yield
estimates biased to the warm/summer seasons in mid- and high-latitude regions (Shanahan et al., 2013;
Foster et al., 2016; Dang et al., 2018), such as in Lake Qinghai (36°54′ N, 100°01′ E) in the
northeastern Tibetan Plateau (Wang et al., 2012), in Lower King pond (44°25′ N, 72°26′ W) in
temperate northern Vermont, U.S.A. (Loomis et al., 2014), and in the Arctic lakes (Peterse et al.,
2014). The warm biased temperature estimates in the mid- and high-latitude lakes have been
postulated to be caused by the higher brGDGT production during warm seasons (e.g., Pearson et al.,
2011; Shanahan et al., 2013).

BrGDGT-producing bacteria in soils could be metabolically active, hence producing abundant

brGDGTs in warm and humid season, but suppressed in cold and/or dry environments (Deng et al.,
2016; De Jonge et al., 2014; Naafs et al., 2017). However, it is presently unclear whether the

brGDGTs in lacustrine sediments are mainly produced during the warm season. Investigations on lake water SPM reveal higher concentration of brGDGTs in the water column may occur in different seasons, e.g., in winter in Lake Lucerne in central Switzerland (Blaga et al., 2011), Lake Challa in tropical Africa (Buckles et al., 2014a) and Lake Huguangyan in subtropical southern China (Hu et al., 2016), in spring and autumn in Lower King Pond in temperate northern Vermont, U.S.A. (Loomis et al., 2014), and in warm season in Lake Donghu in central China (Qian et al., 2019). Moreover, the contribution of the aquatic brGDGTs to the sediments is quantitatively unknown, and likely minor considering that brGDGT producers favor anoxic conditions (Weijers et al., 2006b; Weber et al., 2018) that usually prevail in bottom water and sediments, which may discount the application of SPM-derived findings to the sedimentary brGDGTs.

In fact, brGDGT-based temperature indices should directly record lake water temperature (LWT), rather than AT, if the brGDGTs in lake sediments solely or mainly sourced from the lake environments (Tierney et al., 2010; Loomis et al., 2014). So, the mean annual AT estimate based on lake sedimentary brGDGTs is valid only when LWT is tightly coupled with AT. However, the relationship between LWT and AT is potentially complex in cold regions, as well as in deep lakes, and the coupling between the two is not always the case, which would hamper the application of brGDGTs for temperature estimates (Pearson et al., 2011; Loomis et al., 2014; Weber et al., 2018). In deep lakes, bottom water temperature usually decouples with AT, together with the predominant production of

brGDGTs in deep water and sediments, causing weak correlations between brGDGT-derived

temperature and AT (Weber et al., 2018). For shallow lakes, LWT does not always follow AT either,

specifically in winter when AT is below freezing, in cold regions, as has been shown in the Lower

King pond (Loomis et al., 2014). However, the decoupling between LWT and AT has not been

recognized as a key mechanism for the warm bias of brGDGT-derived temperatures observed widely

in the mid- and high-latitude lakes, and seasonal production or deposition of brGDGTs is usually

invoked as a cause (e.g., Pearson et al., 2011; Shanahan et al., 2013; Loomis et al., 2014). Here, we

hypothesized that the decoupling between LWT and AT in mid- and high-latitude shallow lakes, rather

than the warm season production, could have caused the frequently observed warmer temperature

estimates from the lacustrine brGDGTs. To test this hypothesis, we investigated the Gonghai Lake (a

shallow alpine lake) in north China by collecting SPM and surface sediments in different depths in the

lake and soils in its catchment in a hot summer and a cold winter. We analyzed brGDGT distributions

in these materials to determine the sources of brGDGTs in the lake and further discussed the possible

reasons for the seasonality of brGDGT-estimated temperatures.

## 2 Materials and methods

**2.1 Gonghai Lake**

The Gonghai Lake [38°54′ N, 112°14′ E, ca. 1860 m above sea level (a.s.l.); Fig. 1a and 1b] is

located on a planation surface of the watershed between the Sang-kan River and the Fenhe River at the northeast margin of the Chinese Loess Plateau. The location is close to the northern boundary of the modern East Asian summer monsoon (EASM, Chen et al., 2008; Fig. 1a). The modern local climate is controlled mainly by the East Asian monsoon system, with a relatively warm and humid summer resulting from the prevailing EASM from the southeast, and a relatively cold and arid winter under the prevailing East Asian winter monsoon (EAWM) from the northwest (Chen et al., 2013, 2015; Rao et al., 2016). The mean annual precipitation is ca. 482 mm, concentrated (75%) between July and September (Chen et al., 2013). Its total surface area is ca. 0.36 km$^2$ and the maximum water depth is ca. 10 m. Based on a nearby weather station, the measured mean annual AT is 4.3 °C for the past 30 years. The warm season lasts from May to September (Fig. 1c), when column stratification develops with an upper-bottom temperature difference >1 °C. During the winter from November to March, ice forms on the lake surface, and LWT under ice vertically constant at ca. 4 °C, which is significantly higher than AT that is much below the freezing point (Fig. 1c). From April to October, the ice disappears and LWT follows AT closely, demonstrating a coupling between them (Fig. 1c). The vegetation type of the planation surface belongs to transitional forest-steppe, dominated by *Larix principis-rupprechtii*, *Pinus tabulaeformis* and *Populus davidiana* forest, *Hippophae rhamnoides* scrub, *Bothriochloa ischaemum* grassland and *Carex spp*. (Chen et al., 2013; Shen et al., 2018).

**2.2 Sampling**

In September 2017, five surface soil samples in the catchment and five surface sediment samples at different depths (1.0, 2.5, 5.5, 6.7, 8.0 m) in Gonghai Lake were collected (Fig. 1b). At each soil sample site, we collected 5–6 subsamples (top 0–2 cm) within an area of ca. 100 $m^2$ with contrasting micro-topography or plant cover and then mixed them to represent a single sample. To avoid possible human disturbances, the soil sampling sites were distant from roads and buildings. All samples collected in the field were stored in a refrigeration container during transportation and then freeze-dried for >48 h in the laboratory. Details of all the sampling sites, including locations, sample depth and vegetation type, are listed in Table 1.

In addition, we also collected two batches of SPM samples at water depth of 1 m, 3 m, 6 m and 8 m by filtering 50 L water through a 0.7 μm Whatman GF/F filter on site in September 2017 and January 2018, respectively. SPM samples were also stored in a refrigeration container during transportation and then freeze-dried for >48 h in the laboratory. At the same time of SPM sampling, we measured water column parameters in the lake using an YSI water quality profiler.

**2.3 Sample treatment and GDGT analysis**

Freeze-dried soil and sediment samples were homogenized at room temperature, and accurately weighed. Each freeze-dried filter with SPM attached was cut into small pieces using a sterilized scissor. Each sample of soil, sediment and SPM was placed in a 50 mL tube and then ultra-sonicated successively with dichloromethane/methanol (DCM/MeOH, 1:1, v/v) four times. After centrifugation

and combination of all the extracts of a sample, an internal standard consisting of synthesized $C_{46}$
GDGT was added with a known amount (Huguet et al., 2006). Subsequently, the total extracts were
concentrated using a vacuum rotary evaporator. The nonpolar and polar fractions in the extracts were
separated via silica gel column chromatography, using pure $n$-hexane and DCM/MeOH (1:1, v/v),
respectively. The polar fraction containing GDGTs was dried in a gentle flow of $N_2$, dissolved in
$n$-hexane/ethyl acetate (EtOA) (84:16, v/v) and filtered through a 0.45 μm polytetrafluoroethylene
filter before instrumental analysis. We performed GDGT analysis by high performance liquid
chromatography-atmospheric pressure chemical ionization-mass spectrometry (HPLC-APCI-MS;
Agilent 1200 series 6460 QQQ). Following the method of Yang et al. (2015), the separation of 5- and
6-methyl brGDGTs was achieved using two silica columns in tandem (150 mm × 2.1 mm, 1.9 μm,
Thermo Finnigan; U.S.A.) maintained at 40 °C. The following elution gradient was used: 84/16
$n$-hexane/EtOA (A/B) to 82/18 A/B from 5 to 65 min and then to 100% B in 21 min, followed by
100% B for 4 min to wash the column and then back to 84/16 A/B to equilibrate it for 30 min. The
flow rate was at a constant 0.2 ml/min throughout. BrGDGTs were ionized and detected with single
ion monitoring (SIM) at m/z 1050, 1048, 1046, 1036, 1034, 1032, 1022, 1020, 1018 and 744. The
brGDGTs were quantified from comparing retention time and peak areas with the $C_{46}$ GDGT internal
standard. Based on duplicate HPLC/MS analyses, the analytical errors of both the MBT'$_{5ME}$ and
MBT'$_{6ME}$ index were ±0.01 units.

## 2.4 Calculation of GDGT-related Proxies

The MBT'$_{5ME}$ and MBT'$_{6ME}$ index were calculated following Eq. (1) and (2) as in De Jonge et al. (2014):

$$MBT'_{5ME} = (Ia+Ib+Ic)/ (Ia+Ib+Ic+IIa+IIb+IIc+IIIa) \qquad (1)$$

$$MBT'_{6ME} = (Ia+Ib+Ic)/ (Ia+Ib+Ic+IIa'+IIb'+IIc'+IIIa') \qquad (2)$$

The isomer ratio (IR) of 6-methyl was calculated as in De Jonge et al. (2014). The $\Sigma IIIa/\Sigma IIa$ ratio was calculated as in Martin et al. (2019), which is modified from Xiao et al. (2016). The weighted average number of ring moieties (#Rings$_{tetra}$, #Rings$_{penta\ 5ME}$ and #Rings$_{penta\ 6ME}$) followed Sinninghe Damsté (2016):

$$IR_{6ME} = (IIa'+IIb'+IIc'+IIIa'+IIIb'+IIIc')/ (IIa+IIa'+IIb+IIb'+IIc+IIc'+IIIa+IIIa'+IIIb+$$

$$IIIb'+IIIc+IIIc') \qquad (3)$$

$$\Sigma IIIa/\Sigma IIa = (IIIa+IIIa'+IIIa'')/ (IIa+IIa') \qquad (4)$$

$$\#Rings_{tetra} = (Ic*2+Ib)/ (Ia+Ib+Ic) \qquad (5)$$

$$\#Rings_{penta\ 5ME} = (IIc*2+IIb)/ (IIa+IIb+IIc) \qquad (6)$$

$$\#Rings_{penta\ 6ME} = (IIc'*2+IIb')/ (IIa'+IIb'+IIc') \qquad (7);$$

The Roman numerals represent different brGDGT homologues referred to Yang et al. (2015) and Weber et al. (2015) (see Appendix 1).

In this study, we used two silica columns in tandem and successfully separated 5- and 6-methyl

brGDGTs. However, many previous brGDGT studies on lake materials used one cyano column, which did not separate 5- and 6-methyl brGDGTs (e.g., Wang et al., 2012; Loomis et al., 2014; Hu et al., 2015, 2016; Cao et al., 2017). In order to facilitate comparison with previous studies, we reanalyzed the published brGDGT data without separation of 5- and 6-methyl brGDGTs in the Gonghai Lake (Cao et al., 2017). For temperature estimations, we listed the Eqs. (8–17) used in this study in Table 2.

## 3    Results

### 3.1 Seasonal changes in environmental parameters

The AT in our study area ranged from −12.2 to 21.6 °C, below freezing in winter (November to February) and at 4.3 °C for the mean in the year 2018 (Fig. 1c). Surface LWT ranged from 3.4 to 21.9 °C (average 10.6 °C), and remained stable at ca. 4 °C in winter (Fig. 1c). In September 2017, water column stratification was weak with temperature ranging from 16.9 to 17.8 °C and exhibiting a gradual and slight decrease with depth (Fig. 2). In January 2018, the lake surface water was frozen and LWTs under ice were 4 °C at all depths (Fig. 2).

### 3.2 Concentration and distribution of brGDGTs

BrGDGTs were detected in all samples, and their total concentration ranged between 16–75 ng/g dry weight (dw) in surface soils from Gonghai catchment, 42–707 ng/g dw in lake surface sediments,

5–10 ng/l in September and 3–8 ng/l in January in water SPM (Table 1 and Fig. 2). The average
content of brGDGTs in lake surface sediments (291 ng/g dw) was significantly higher than in surface
soils (31 ng/g dw) and particularly exhibited an increasing trend with water depth. In SPM, the
average concentration of brGDGTs in water column showed no significant difference between
September and January (t = 1.2, $p$ = 0.26) but there was a clearer trend of increase with depth in
September than in January (Fig. 2). Notably, the compound IIIa", which was regarded typical for in
situ produced lacustrine brGDGTs (Weber et al., 2015), was also identified in the Gonghai Lake
sediments and SPM but not found in catchment soils (Table1 and Fig. 3a). There was no significant
difference in average concentration of IIIa" in water column between September and January (t = 0.62,
$p$ = 0.28). The change patterns of IIIa" with water depth in SPM and sediments were the same as those
of the total brGDGTs (Table 1).
The brGDGTs in soils, sediments and SPM were dominated by brGDGT II and III series, with
acyclic compounds dominant in every series (Fig. 3a). In comparison, the mean ΣIIIa/ΣIIa ratio value
in sediments (1.14–1.52 range, 1.30 average) was higher than in SPM (0.84–1.11 range, 0.99 average)
and soils (0.56–0.86 range, 0.70 average). In addition, 6-methyl brGDGTs dominated over 5-methyl
brGDGTs in soils, exhibiting mean $IR_{6ME}$ of 0.62; whereas the two isomers were similar in content in
sediments ($IR_{6ME}$ = 0.47–0.60 range, 0.51 average) and SPM ($IR_{6ME}$ = 0.45–0.50 range, 0.48 average)
(Fig. 3a).

### 3.3 Cyclisation ratio, methylation index of brGDGTs

The #Rings$_{tetra}$ values varied from 0.26 to 0.45 (0.36 average) in catchment soils, 0.37–0.43 (0.40 average) in September and 0.39–0.42 (0.40 average) in January in SPM, and 0.45–0.47 (0.45 average) in surface sediments (Fig. 3b). The #Rings$_{penta\ 5ME}$ showed the same increasing trend as #Rings$_{tetra}$ from soils to SPM and then to sediments (Fig. 3b). In contrast, #Rings$_{penta\ 6ME}$ in soils was similar to that in sediments and SPM (Fig. 3b).

The MBT'$_{5ME}$ values varied from 0.31 to 0.36 (average 0.35) in catchment soils, 0.23–0.29 (0.26 average) in surface sediments, 0.23–0.28 (0.26 average) in September and 0.24–0.26 (0.25 average) in January in SPM (Fig. 3b). Generally, the MBT'$_{5ME}$ exhibited decreasing trends with water depth in surface sediments and SPM in September (Fig. 2). The MBT'$_{6ME}$ values varied from 0.20 to 0.33 (0.25 average) in surface soils of the lake catchment, 0.22–0.27 (0.25 average) in surface sediments, 0.24–0.32 (0.28 average) in September and 0.26–0.28 (0.27 average) in January in SPM (Fig. 3b). The MBT'$_{6ME}$ also decreased in SPM in September, but increased in sediments with water depth. Both MBT'$_{5ME}$ and MBT'$_{6ME}$ changed less in SPM in January with water depth (Fig. 2).

## 4   Discussions

### 4.1 In situ production of brGDGTs in the Gonghai Lake

Although brGDGTs have a strong potential to record temperature in lacustrine regions (Tierney

et al., 2010; Pearson et al., 2011; Sun et al., 2011; Loomis et al., 2012; Dang et al., 2018; Russell et al., 2018), the sources of brGDGTs in lake sediments should be carefully identified. There are two potential sources, including allochthonous input from soil and autochthonous production in lake water and/or surface sediments, which can be distinguished by comparison of brGDGT distribution between surface sediments and soils (Tierney and Russell, 2009; Loomis et al., 2011; Wang et al., 2012; Hu et al., 2015; Sinninghe Damsté, 2016).

In the Gonghai Lake, the average content of brGDGTs in surface sediments was significantly higher than that in surface soils (Table 1). Moreover, they exhibited a clearly increasing trend with water depth, suggesting a possible autochthonous contribution, even though soil brGDGTs input cannot be ignored. Moreover, the brGDGT distribution in surface sediments was similar to that of SPM, but quite different from that of soils (Fig. 3a). Several lines of evidence indicate a substantial in situ production of brGDGTs in the Gonghai Lake. (I) The presence of IIIa″ in the Gonghai Lake sediments and SPM but the absence in the catchment soils may be a direct evidence of in situ production in the lake (Fig. 3a). A similar conclusion has been drawn in a Swiss mountain lake basin (Weber et al., 2015). (II) In the Gonghai Lake, the ΣIIIa/ΣIIa ratio in sediments (1.3 average) and SPM (0.99 average) were much higher than in catchment soils (0.7 average) (Fig. 3a). The values of ΣIIIa/ΣIIa >0.92 has been regarded as the evidence of aquatic production in previous reports (Xiao et al., 2016; Martin et al., 2019; Zhang et al., 2020). (III) The average values of $IR_{6ME}$ in surface

sediments and SPM were significantly lower than in catchment soils (Fig. 3a), suggesting at least
some of 5-methyl brGDGTs in lake sediments and SPM were produced in situ. (IV) The cyclisation
ratio of brGDGTs has been also used to distinguish the aquatic production, although applied to marine
sediments, from soil input (Sinninghe Damsté, 2016). In the Gonghai Lake, #Rings$_{tetra}$ and #Rings$_{penta}$
$_{5ME}$ were clearly higher in sediments than in catchment soils ($p$ <0.05 for #Rings$_{tetra}$, $p$ <0.01 for
#Rings$_{penta\ 5ME}$), although #Rings$_{penta\ 6ME}$ in sediments was similar to that in catchment soils ($p$ = 0.11
for #Rings$_{penta\ 6ME}$; Fig. 3b).

The in situ production of brGDGTs in the Gonghai Lake can be also evidenced by the

discrepancies in reconstructed temperatures between soils and sediments/SPM. Based on the new
global soil calibration of Eq. (9) and regional soil calibration of Eq. (10) for China, the
brGDGT-derived AT in the Gonghai catchment soils ranged from 1.18 to 2.75 °C (average 2.33 ±
0.65 °C; Table 1) and from −4.22 to −1.21 °C (average −2.42 ± 1.19 °C; Table 1), respectively.
Considering the ±4.8 °C uncertainty of the global calibration and ±2.5 °C of the regional calibration,
the estimated temperatures from the global calibration are much close to the mean annual AT of
4.3 °C, thereby well reflecting mean annual AT in our study lake catchment. Then, the global
calibration Eq. (9) was applied to sediment/SPM data, yielding estimated temperatures −0.50 ±
0.78 °C in surface sediments and −0.55 ± 0.52 °C in SPM and hence much lower than those from
surface soils (2.33 ± 0.65 °C; Table 1). Similarly, temperature underestimation using soil-derived
calibration has been widely reported in many modern lake sediments (e.g., Tierney et al., 2010;
Loomis et al., 2012; Pearson et al., 2011; Russell et al., 2018), which has been attributed to in situ
production of brGDGTs in the lakes.
**4.2 Lacustrine brGDGT-derived ATs are warm season biased (average monthly**
**temperature >0 °C)**

The suggested in situ production of brGDGTs prompts us to use lake-specific temperature

calibrations (Tierney et al., 2010; Pearson et al., 2011; Sun et al., 2011; Loomis et al., 2012; Dang et
al., 2018; Russell et al., 2018) to reconstruct AT, although not differentiated quantitatively the relative
contributions of aquatic vs. soil-derived brGDGTs. Here, we applied four equations, Eqs. (11) and
(15)–(17) in Table 2, to our sedimentary brGDGT data.

As shown in Fig. 4a, the reconstructed temperatures using different equations are >6.4 °C.

Despite discrepancies in the temperature values between calibrations, they are comparable
considering the uncertainty of each calibration. A prominent feature of the reconstructed temperature
is that they, especially those in the shallower sediments, are well above the annual mean AT but more
close to the mean warm season AT (average monthly temperature >0 °C). This feature is consistent
with numerous studies proposing that lacustrine brGDGT-derived ATs are warm season biased
(Shanahan et al., 2013; Peterse et al., 2014; Dang et al., 2018).

Many previous brGDGT instrumental analyses on lake materials used one cyano column, which

did not separate 5- and 6-methyl brGDGTs. Using the data published in the same lake from Cao et al.

(2017), we re-calculated temperature using different calibrations. The results showed that the absolute

temperature estimates were all significantly warmer than the mean annual AT (Table 3), with the

temperature offsets varying from 4–10 °C, which cannot be fully explained by the uncertainty of each

calibration. Therefore, it appears that sedimentary brGDGT-derived temperature is warm season

biased in the Gonghai Lake irrespective of whether or not 5- and 6-methyl brGDGTs are separated.

Moreover, we found the warm season bias of reconstructed AT is increasingly apparent with the

increase of latitude. Here, five lakes, including Lower King pond (Loomis et al., 2014), Qinghai Lake

(Wang et al., 2012), Lake Donghu (Qian et al., 2019), Huguangyan maar (Hu et al., 2015, 2016) and

Lake Towuli (Tierney and Russell, 2009), were selected to compare as an example. These lakes are

located in different regions spanning a relatively large environmental gradient, and more importantly,

brGDGT data from both the lake surface sediments and the surrounding soils are available. We

re-calculated temperatures from published data of brGDGTs from these lakes (Fig. 5) by applying the

calibration of global soils (Eq (8); Peterse et al., 2012) to the surrounding soils and the calibration of

lake surface sediments (Eq (11); Sun et al., 2011) to the lake sediments. As shown in Fig. 5a, the

brGDGT-inferred temperatures in catchment soils are similar to local mean annual ATs. In contrast,

the brGDGT-inferred temperatures in lake sediments are similar to the local mean annual ATs only in

low-latitude lakes, whereas they become increasingly higher than the local mean annual ATs toward

higher latitudes (Fig. 5b). In comparison, the brGDGT-inferred temperatures are close to the local
mean ATs in warm season (average monthly mean AT >0°C) in all these lakes (Fig. 5c). Besides
above discussed lakes, some investigations have also pointed out that brGDGT-inferred temperatures
are higher than mean annual AT, close to warm season AT or summer AT in mid- and high-latitude
lakes (Shanahan et al., 2013; Peterse et al., 2014; Foster et al., 2016; Dang et al., 2018), but close to,
or lower than, mean annual AT in low-latitude lakes (Tierney et al., 2010; Loomis et al., 2012).
Therefore, it is a global occurrence that sedimentary brGDGT-derived temperatures are warm season
biased in lakes at cold regions.
4.3. **Lacustrine brGDGTs reflect deep/bottom water temperature**
Another feature of sedimentary brGDGT-derived ATs in our results is that there is a consistently
decreasing trend of reconstructed temperature with depth using Eqs. (11), (15) and (16) (Fig. 4a),
albeit less clear using Eq. (15). It is not understandable that AT is correlated with water depth.
Interestingly, both MBT'$_{5ME}$ and MBT'$_{6ME}$ in SPM showed decreasing trends with water depth in
September, similar to the water temperature profile of the month (Fig. 2). In January, the relatively
unchanged MBT'$_{5ME}$ and MBT'$_{6ME}$ (<0.02) also mirror the constant water temperature of the month
(Fig. 2). Accordingly, we surmise that brGDGT-derived temperatures in sediments and SPM may
actually reflect water temperature.
Although the MBT'$_{5ME}$ and MBT'$_{6ME}$ in SPM in the lake seem to reflect temperature changes in

the water column to some extent, the differences of brGDGT-derived temperatures based on

lake-specific calibrations between September and January (−0.93 to 1.21 °C) are much lower than the

measured difference (~13 °C), independent of the calibration of (15), (16) or (17) (Tables 1 and 2). In

fact, similar results have been also reported in other lakes. For example, in the Lower King Pond, the

calculated seasonal temperature difference in surface water SPM was 5.4 °C, significantly smaller

than the measured difference about 28.3 °C (Loomis et al., 2014); in the Huguangyan maar lake, the

calculated seasonal temperature difference was 8 °C, also significantly smaller than the measured

difference about 16 °C (Hu et al., 2016). The reduced seasonal contrasts in SPM brGDGT-derived

temperatures could result from the existence of "fossil" brGDGTs and sediment resuspension in the

water column, which may lead to a long (e.g., multi seasonal) residence time of SPM, although not

exactly known (Loomis et al., 2014). The even smaller differences in MBT′$_{5ME}$ and MBT′$_{6ME}$ between

sediments and SPM at deeper sites in our results (Fig. 2) suggest the impacts of sediment suspension

on SPM. Such a scenario may lead to more "fossil" brGDGTs in SPM than those produced within a

specific season or month, as evidenced by an observation showing that only a small proportion of

intact polar lipid of brGDGTs, indicative of fresh brGDGTs, was detected in total brGDGTs in SPM

in a shallow lake (Qian et al., 2019). Besides, several parameters, such as ΣIIIa/ΣIIa, IR$_{6ME}$, #Rings$_{tetra}$

and #Rings$_{penta}$ in SPM were in-between the soil and sediment values, we speculate terrestrial inputs

may be a factor, if any, to reduce the seasonal changes of brGDGTs in SPM.

In addition to reflecting water temperature, the decease trend with depth in sedimentary
brGDGT-derived temperature further suggests a controlling influence of deep/bottom water
temperature. Similar occurrence has been observed also in Lower King pond in temperate northern
Vermont, U.S.A. and Lake Biwa in central Japan, showing that the sedimentary brGDGT-derived
temperatures decreased with water depth, co-varied with mean annual LWT at depths (Ajiako et al.,
2014; Loomis et al., 2014). Also in Loch Lomond in the UK, the brGDGT-derived temperatures by
different MBT/CBT lacustrine calibrations all decreased with water depth (Buckles et al., 2014b). So,
a water depth-related production of brGDGTs should be considered when interpreting
brGDGT-derived temperatures, which will be discussed below.
We notice recent works suggesting that changes in microbial community composition may be
responsible for variations in the distribution of brGDGTs, causing the different responses of soil
brGDGTs temperature, as well as pH, under different temperature ranges (e.g., De Jonge et al. 2019).
However, little is known about whether this idea is applicable to aquatic environments. According to
De Jonge et al. (2019), community change can be indicated by the community index (CI =
Community Index) in soils, with CI >0.64 indicating warm community cluster and CI <0.64
indicating cold community cluster. Here we applied the CI to lake sediment data including ours and
those available for the entire 15 brGDGT compounds in literature, mostly from the east Africa. As
shown in Fig. 4b, the putative two community clusters also occur in lake environments, with the

Gonghai community belonging to the "cold" cluster. Different from soil data showing that MBT'$_{5ME}$ captures large temperature changes only when the bacterial community shows a strong change in composition (De Jonge et al. 2019), it seems that MBT'$_{5ME}$ changes linearly with LWT, which is less influenced by the bacterial community change (Fig. 4b). However, we note that the test of community change here is rather crude, and further studies on the biological sources of brGDGT and their response to temperature in aquatic environments are needed.

**4.4 Ice cover formation as a mechanism for the apparent warm bias of lacustrine brGDGT-derived temperature**

One explanation for the warm season biases of the lacustrine brGDGT-derived temperature in mid to high latitudes has been proposed as the excessive production of brGDGTs during the warm/summer season relative to winter season (Pearson et al., 2011; Shanahan et al., 2013; Peterse et al., 2014; Foster et al., 2016; Dang et al., 2018). In the Gonghai Lake, the average concentration of brGDGTs in SPM is $7.1 \pm 2.0$ ng/l in September and $5.2 \pm 2.3$ ng/l in January (Fig. 2) with no significant difference. Besides, the compound IIIa", which is likely specifically of aquatic origin (Weber et al., 2015), also showed no significant seasonal difference ($0.36 \pm 0.09$ ng/l in September vs. $0.31 \pm 0.15$ ng/l in January). More importantly, the small differences in MBT'$_{5ME}$ and MBT'$_{6ME}$ of SPM and their derived temperatures between September and January suggest that the actual seasonal temperature difference, which may be recorded by the immediately produced brGDGTs, would have

been substantially masked or smoothed by the predominance of fossil brGDGTs. In addition, brGDGT-derived temperatures in SPM were close to mean annual water temperature and lower than the mean annual warm water temperature, also did not support the excessive production of brGDGTs during the warm/summer season relative to winter season. Besides, the season of higher brGDGT concentration has been found different in different lakes, e.g., in spring and autumn in Lower King pond (Loomis et al., 2014), in winter in Lake Lucerne (Blaga et al., 2011), and in summer in Lake Donghu in central China (Qian et al., 2019). However, in all these lakes in temperate climate zones, the brGDGT-derived temperatures have been found to be slightly or significantly warm season biased (Loomis et al., 2014; Qian et al., 2019; Fig. 5b). The above evidence suggests that other factors, other than seasonality in the production of brGDGTs in the lakes, should be responsible for the bias of brGDGT-inferred temperature toward warm season in higher latitudes (Fig. 5b and c).

The brGDGT-derived temperature in lake sediments could be influenced by the vertically inhomogeneous production of brGDGTs with maximum in deep/bottom waters. This seems true in the Gonghai Lake as evidenced by the increase of sedimentary brGDGT content and the decrease of brGDGT-derived temperature with water depth as discussed above. The bio-precursors of brGDGTs have been proposed to be bacteria with an anaerobic heterotrophic lifestyle (Sinninghe Damsté et al., 2000; Weijers et al., 2006b, 2010; Weber et al., 2015, 2018), implying that a potentially anoxic (micro)environment in deep/bottom water favors the production of brGDGTs (Woltering et al., 2012;

Zhang et al., 2016; Weber et al., 2018). Such an occurrence could lead to higher proportion of 'colder temperature' brGDGTs in lake sediments, which may at least partly interpret the frequently observed cool bias of brGDGT-derived temperatures in many lakes, such as the Lake Challa, Lake Albert, Lake Edward and Lake Tanganyika (Tierney et al., 2010; Loomis et al., 2012; Buckles et al., 2014a). The MBT/CBT-derived temperature in the tropical Lake Huguangyan was thought to reflect mean annual AT (Hu et al., 2015, 2016); however, has recently been proposed to be winter/cool biased (Chu et al., 2017). We suppose that, as a monomictic lake, the lower mean annual temperature than mean annual AT in deep/bottom waters might be a cause for the cool biased brGDGT temperature in the lake. Intriguingly, all the above lakes are in the tropics. Nonetheless, the deep/bottom water bias may be still true for the brGDGT-derived temperature in lakes in higher latitude, as suggested by our data in the Gonghai Lake. However, different from those tropical lakes, in higher-latitude lakes, including the Gonghai Lake (this study), Qinghai Lake (Wang et al., 2012), Lower King pond (Loomis et al., 2014), some cold-region lakes in China (Dang et al., 2018) and some Arctic lakes (Shanahan et al., 2013; Peterse et al., 2014), the sedimentary brGDGT-derived temperatures are all higher, not lower, than the mean annual AT. Therefore, more production of brGDGTs in deep/bottom water alone is not responsible for the warm bias of brGDGT-derived temperature in surface sediments at least in these lakes.

Although brGDGTs in lake sediments were confirmed to be mainly derived from in situ aquatic

production, previous studies deemed that the estimated temperatures can still reflect AT by assuming
that LWT is tightly coupled with AT (Tierney et al., 2010). In fact, such tight coupling can be found in
tropical-subtropical lakes, where AT is always above the freezing point, but is not true in
higher-latitude lakes such as Lower King pond and Gonghai Lake with lake surface freezing in winter
(Fig. 6a and b). The reason is that lake surface ice prevents the thermal exchange between water and
air, leading to decoupling between LWT (usually $\geq 4$ °C) and AT (<0 °C) in winter in cold regions. The
decoupling makes mean annual LWT, even at the deep/bottom waters, higher than mean annual AT.
Therefore, the greater warm biases of brGDGT-derived temperatures from surface sediments in higher
latitudes (Fig. 5b) could be due to the stronger decoupling (e.g., longer freezing time) between LWT
and AT. Nevertheless, annual mean LWT appears basically close to the mean AT in warm season
(average monthly temperature >0 °C) (Fig. 6f), which could be the reason why the brGDGT-inferred
temperatures are similar to the mean warm season AT (Fig. 5c). Due to lack of detailed AT and LWT
data in literature, we failed to show more examples than as shown in Fig. 6, especially those from
even higher latitudes. However, we proposed a simple model for the relationship between LWT and
AT in a year cycle (Fig. 7), which may be a universal physical phenomenon in shallow lakes. In the
mid- and high-latitude region, we believe the decoupling between AT and LWT caused by ice
formation in winter may be applied to explain the observed seasonality of the brGDGT temperature
records. For example, the biases of brGDGT-derived temperatures toward summer AT observed
extensively in the Arctic and Antarctic lakes (Shanahan et al., 2013; Foster et al., 2016) are
compatible with the mechanism that we propose here. Of course, considering limited data in this study,
more investigations are needed to test our viewpoint in future studies.

**5    Conclusions**
We investigated the brGDGT distribution in catchment soils, surface sediments and water
column SPM in September and January in the Gonghai Lake in north China. The lake is characterized
by ice formation on its surface and a constant 4 °C condition in the underlying water in winter. The
brGDGT distribution in sediments were similar to that in SPM but differed clearly from that in soils,
indicating mainly in situ production of brGDGTs in the lake. BrGDGTs in SPM showed little seasonal
differences in concentration and MBT'$_{5ME}$, likely due to a dominant contribution of fossil brGDGTs
caused by, e.g., sediment suspension, which may mask any seasonal signals documented in
sedimentary brGDGTs. The increase of brGDGT content and decrease of methylation index with
water depth in sediments suggested more contribution of aquatic brGDGTs produced from
deep/bottom waters. Based on available lake calibrations, we found that the temperature estimates in
surface sediments and SPM of the Gonghai Lake were higher than the measured mean annual AT but
close to warm season AT, which cannot interpreted by more aquatic production of brGDGTs in warm
season and/or in deep/bottom waters. We found that such a warm biased brGDGT-derived temperature
was actually close to the mean annual LWT, and therefore proposed that water-air temperature
decoupling due to ice formation at the lake surface in winter, which can prevent thermal exchange
between lake water and air, may be the cause for the apparent bias toward warm AT of lacustrine
brGDGT-derived temperatures. Since the warm AT bias of brGDGT estimates has been observed
extensively in mid- and high-latitude shallow lakes, we believe the mechanism proposed here could
also be applicable to these lakes.

**Data availability**
The raw data of this study can be accessed from https://figshare.com/s/a4f324247ecd9d1ac575.
**Author contribution**
ZR designed experiments, FS and JC collected samples and JC carried experiments out. JC, GJ and
ZR prepared the manuscript with contributions from all co-authors.
**Conflicts of interest**
The authors declare that they have no conflict of interest.

**Acknowledgments**
The work was supported by the Hunan Provincial Natural Science foundation of China (2018JJ1017),
the National Natural Science Foundation of China (41772373), and the Fundamental Research Funds
for the Central Universities of China (grant no. lzujbky-2018-it77). Two anonymous reviewers are
thanked for their valuable comments.

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

**Captions for Tables and Figures:**

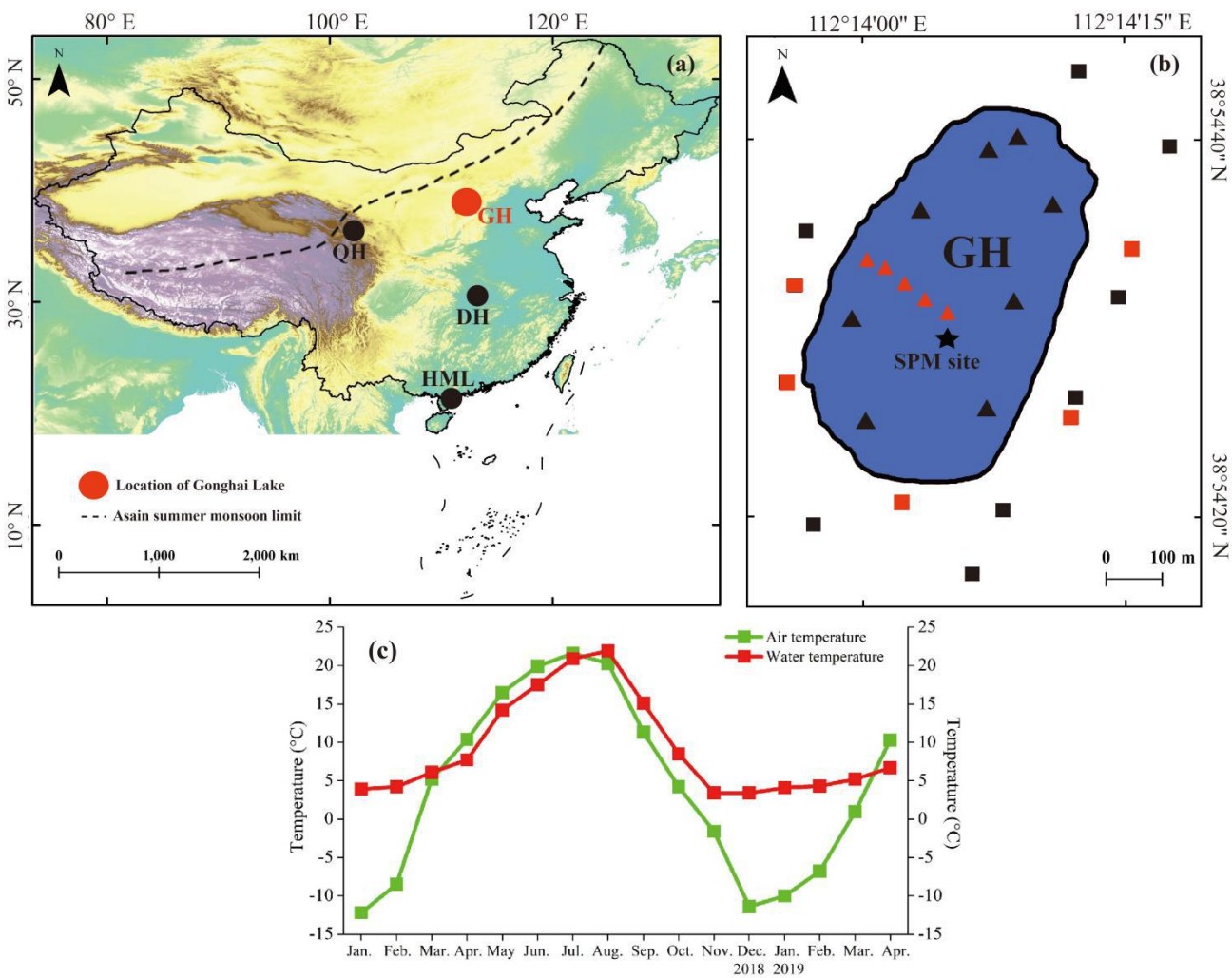


**Fig. 1.** (a) The Gonghai Lake (red circle), other referenced lakes (black circles) and modern Asian summer monsoon

limit (dashed line; Chen et al., 2008). (b) SPM from water column (black star), surface soils (red squares) and

surface sediments (red triangles) in Gonghai Lake in this study; black squares and triangles represents the

sample sites published in Cao et al. (2017) (modified from Cao et al., 2017). (c) Measured local air temperature

(AT) and lake water temperature (LWT) during 2018–2019 (this study).

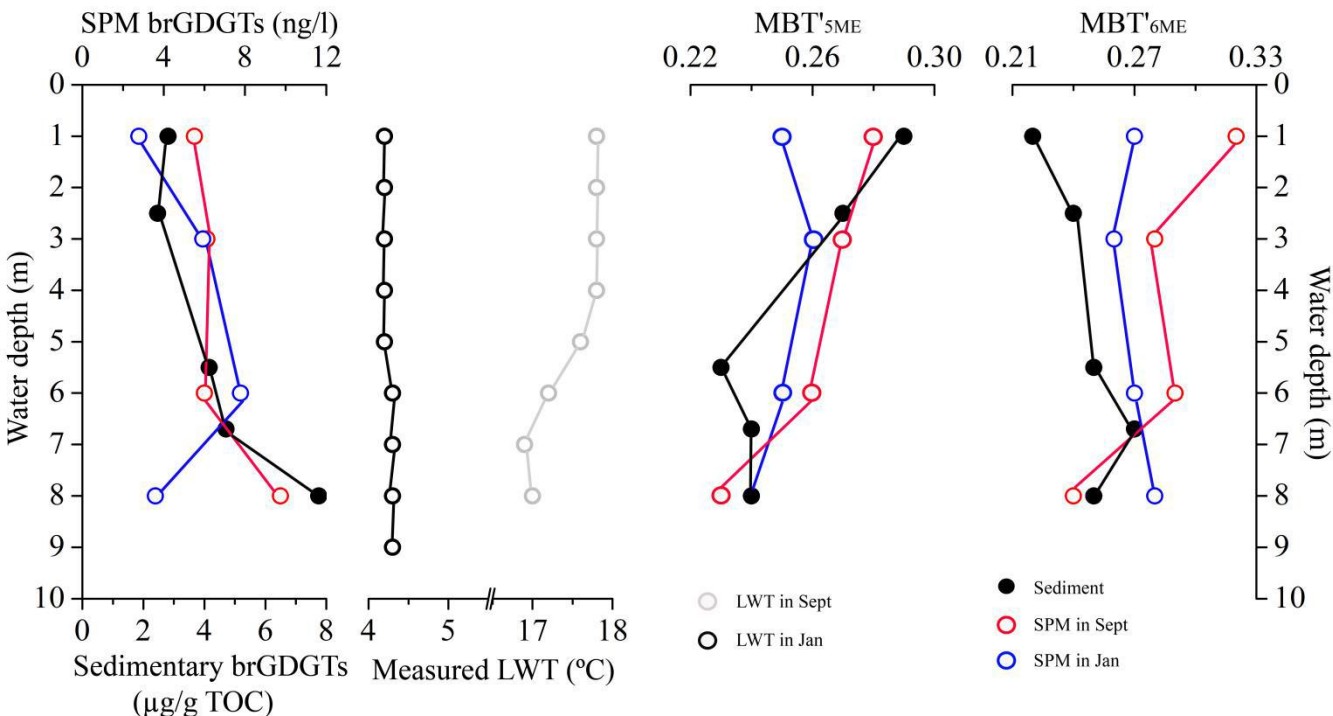


**Fig. 2.** Depth profiles of water temperature, brGDGT concentrations, MBT'$_{5ME}$, MBT'$_{6ME}$ in water SPM from
January and September and sediments in the Gonghai Lake.

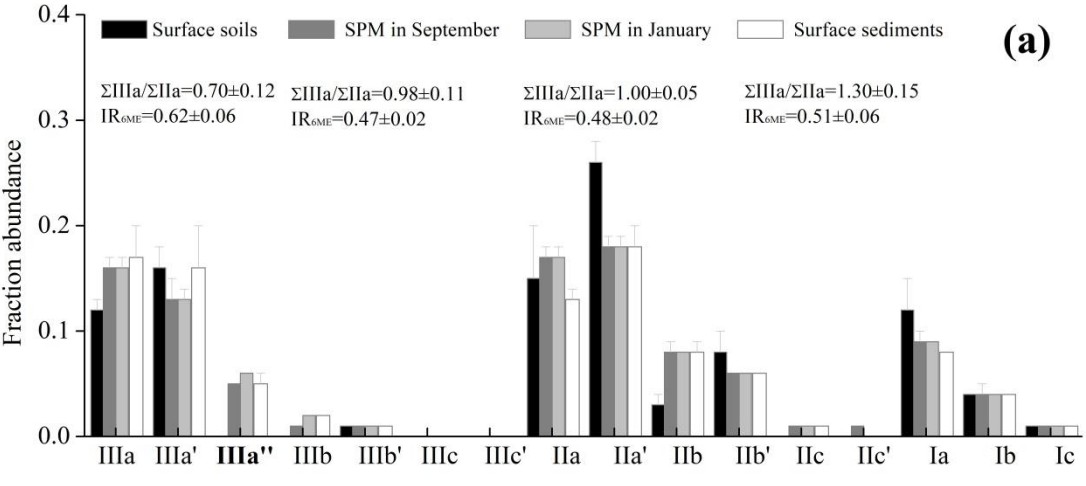

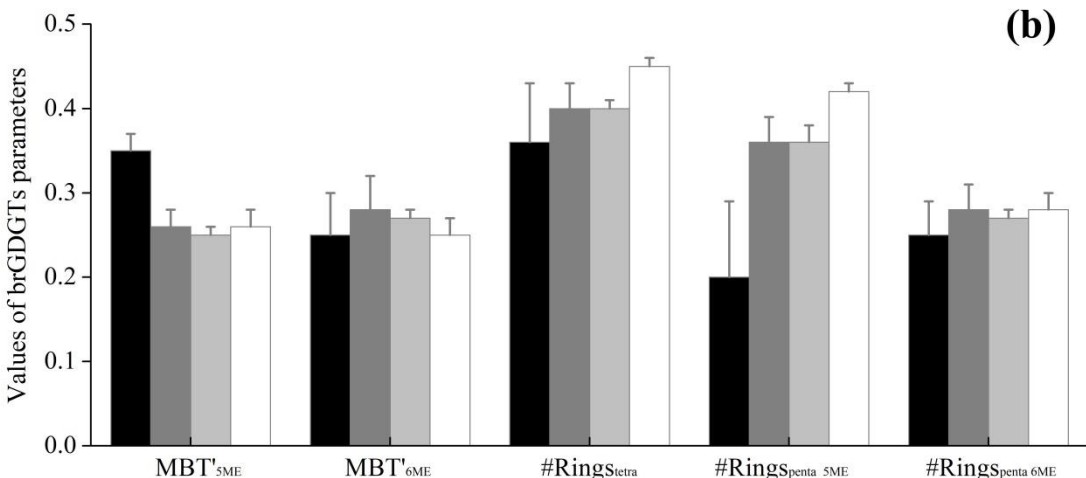


**Fig. 3.** BrGDGT distribution in surface soils, water column (SPM) and surface sediments of the Gonghai Lake. (a)
Fracional abundance of brGDGTs. (b) Degree of methylation and cyclisation of brGDGTs.

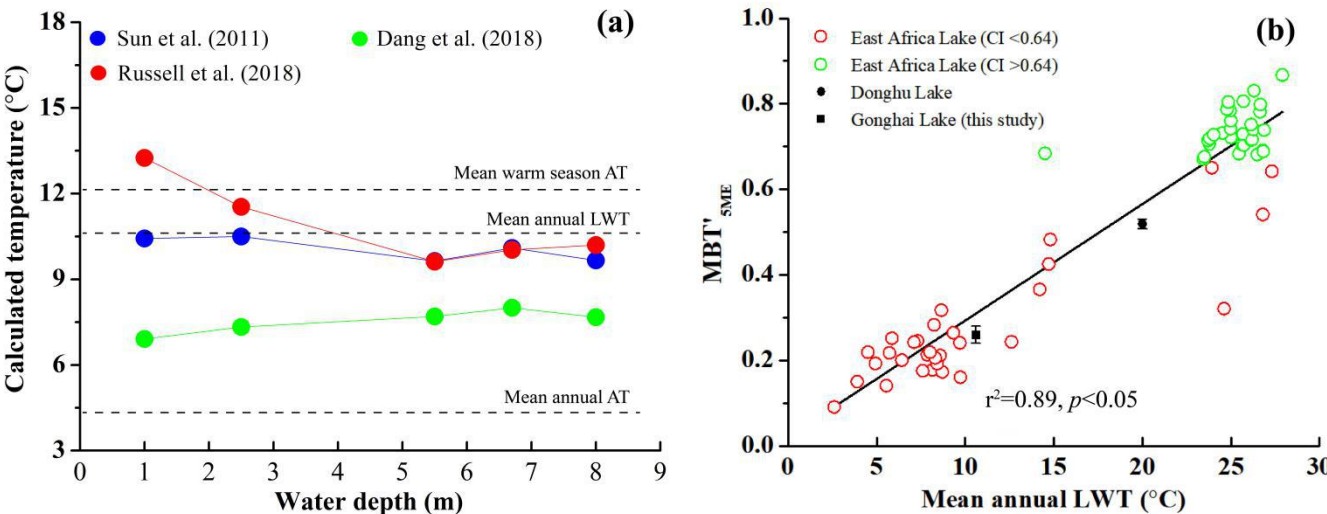


**Fig.** 4. (a) BrGDGT-derived temperatures for sediments using lake calibrations Eqs. (11), (15) and (16) from Sun et
al. (2011), Dang et al. (2018) and Russell et al. (2018) respectively. (b) The correlation between MBT'$_{5ME}$ of
sedimentary brGDGTs and mean annual lake water temperature (LWT); CI index represents Community Index
(De Jonge et al., 2019); the brGDGT data of East Africa Lake, Donghu Lake and Gonghai Lake were sourced
from Russell et al. (2018), Qian et al. (2019) and this study.

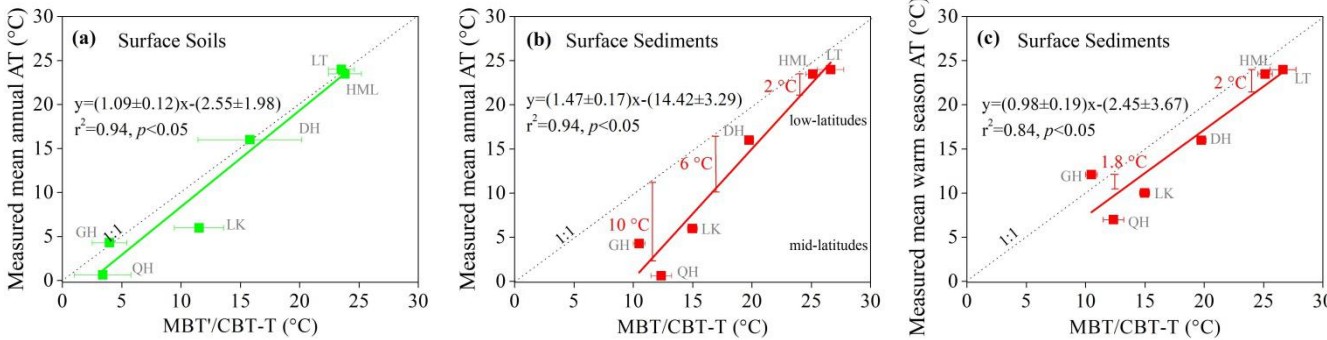

**Fig. 5.** Comparison of brGDGT-derived temperature and measured air temperature. (a) Measured mean annual AT and estimated temperatures of brGDGTs in surface soils based on soil calibration Eq. (9). (b) Measured mean annual AT and estimated temperatures of brGDGTs in surface sediments based on lake calibration Eq. (11). (c) Measured mean warm season AT and estimated temperatures of brGDGTs in surface sediments based on lake calibration Eq. (11). Data are from Gonghai Lake (GH; Cao et al., 2017), Lower King pond (LK; Loomis et al., 2014), Huguangyan maar (HML; Hu et al., 2015, 2016), Lake Donghu (DH; Qian et al., 2019), Qinghai Lake (QH; Wang et al., 2012) and Lake Towuli (LT; Tierney and Russell, 2009).

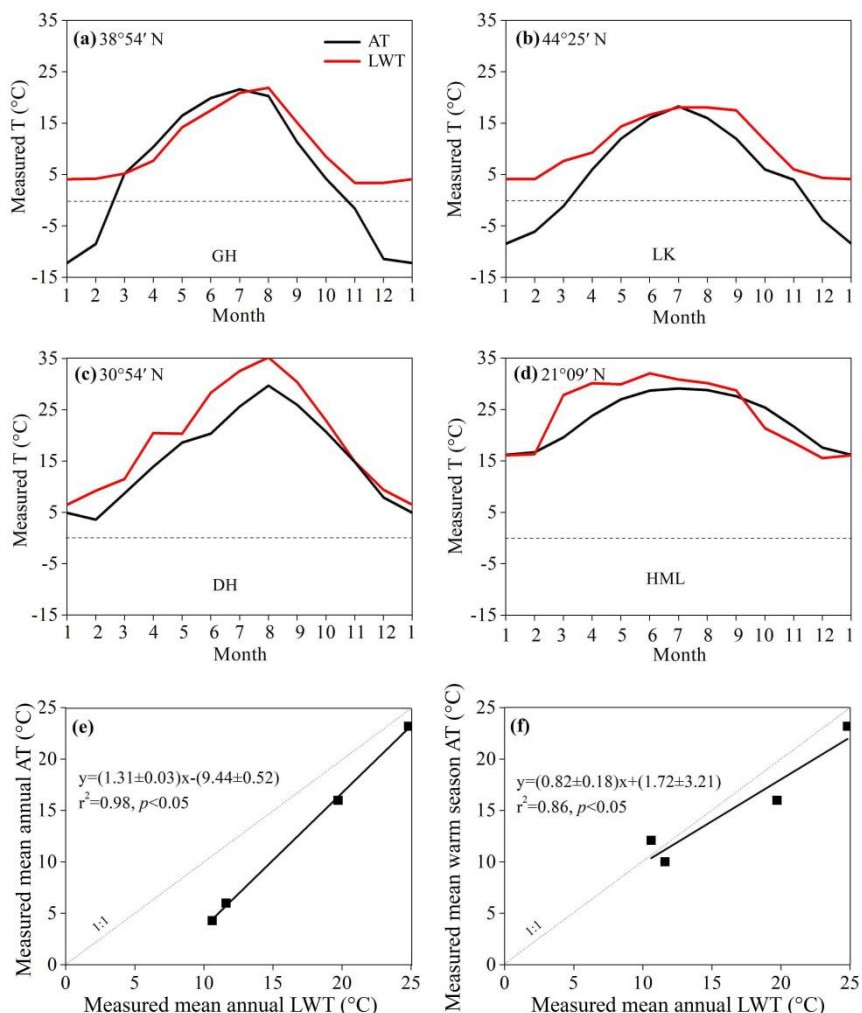

**Fig. 6.** Measured LWT and AT in (a) Gonghai Lake (GH; this study), (b) Lower King pond (LK; modified from

Loomis et al., 2014), (c) Lake Donghu (DH; modified from Qian et al., 2019) and (d) Lake Huguangyan (HML;

modified from Hu et al., 2016). (e) Correlation between mean annual AT and mean annual LWT. (f) Correlation

between mean warm season AT and mean annual LWT. In the mid-latitude Gonghai Lake and Lower King pond,

the surface LWT follows AT only when the AT is above freezing. In the low-latitude Lake Donghu and Lake

Huguangyan, the surface LWT follows AT for the whole year.

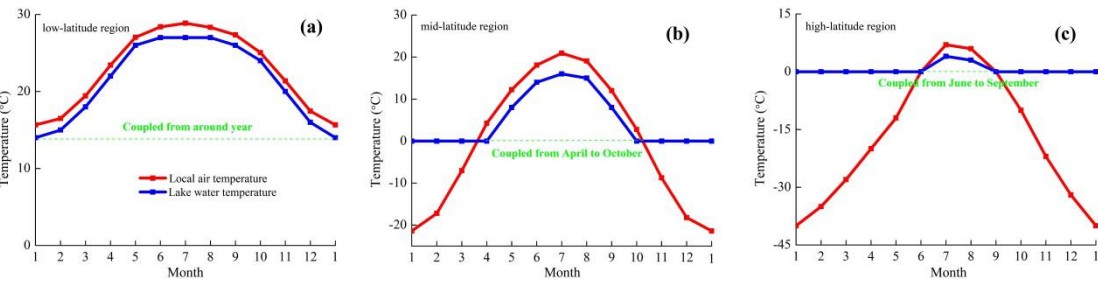

**Fig. 7.** A simple model showing the relationship between LWT and AT in different latitudes

**Table 1** Concentration of brGDGTs, MBT'$_{5ME}$, MBT'$_{6ME}$ and estimated temperatures in catchment surface soils, sediments and water column SPM in the Gonghai Lake.

| Code of site | Longtitude (E) | Latitude (N) | Vegetation type | Water depth (m) | IIIa" (ng/L) | Total brGDGTs (ng/g dw) | MBT'$_{5ME}$ | MBT'$_{6ME}$ | MAAT [a] (°C) | MAAT [b] (°C) | MAAT [c] (°C) | MAAT [d] (°C) | Growth AT [e] (°C) |
|---|---|---|---|---|---|---|---|---|---|---|---|---|---|
| **Surface soils in Gonghai catchment** | | | | | | | | | | | | | |
| S1 | 112° 14'19.039" | 38° 54'37.343" | grass | | 0 | 74.82 | 0.31 | 0.21 | 1.20 | -2.90 | | | |
| S2 | 112° 14'18.460" | 38° 54'28.750" | grass | | 0 | 23.50 | 0.36 | 0.20 | 2.58 | -1.21 | | | |
| S3 | 112° 14'24.140" | 38° 54'23.098" | shrub | | 0 | 22.00 | 0.35 | 0.33 | 2.40 | -4.22 | | | |
| S4 | 112° 14'36.827" | 38° 54'27.126" | shrub | | 0 | 32.65 | 0.36 | 0.26 | 2.64 | -2.15 | | | |
| S5 | 112° 14'40.502" | 38° 54'38.174" | grass | | 0 | 16.06 | 0.36 | 0.24 | 2.82 | -1.61 | | | |
| **Gonghai surface sediments** | | | | | | | | | | | | | |
| D1 | 112° 14'22.963" | 38° 54'36.357" | | 1.00 | 1.46 | 42.03 | 0.29 | 0.22 | 0.70 | -4.24 | 8.35 | 13.50 | 6.91 |
| D2 | 112° 14'24.004" | 38° 54'35.903" | | 2.50 | 1.59 | 33.95 | 0.27 | 0.24 | -0.13 | -4.79 | 7.50 | 11.91 | 7.33 |
| D3 | 112° 14'25.109" | 38° 54'35.294" | | 5.50 | 17.87 | 327.62 | 0.23 | 0.25 | -1.19 | -6.53 | 6.40 | 10.11 | 7.70 |
| D4 | 112° 14'27.301" | 38° 54'34.499" | | 6.70 | 25.53 | 374.29 | 0.24 | 0.27 | -0.93 | -7.32 | 6.67 | 10.57 | 8.00 |
| D5 | 112° 14'28.453" | 38° 54'33.980" | | 8.00 | 42.96 | 706.72 | 0.24 | 0.25 | -0.95 | -6.44 | 6.64 | 10.72 | 7.67 |
| **Gonghai SPM in Sept** | | | | | | | | | | | | | |
| Water-1 m | 112° 14'28.453" | 38° 54'33.980" | | 1.00 | 0.29 | 5.71 | 0.28 | 0.32 | 0.24 | -6.00 | 7.88 | 11.19 | 9.16 |
| Water-3 m | 112° 14'28.453" | 38° 54'33.980" | | 3.00 | 0.36 | 6.39 | 0.27 | 0.28 | -0.05 | -5.46 | 7.57 | 10.86 | 8.25 |
| Water-6 m | 112° 14'28.453" | 38° 54'33.980" | | 6.00 | 0.30 | 6.22 | 0.26 | 0.29 | -0.35 | -6.55 | 7.26 | 10.45 | 8.55 |
| Water-8 m | 112° 14'28.453" | 38° 54'33.980" | | 8.00 | 0.49 | 10.07 | 0.23 | 0.24 | -1.40 | -6.79 | 6.18 | 10.60 | 7.31 |
| **Gonghai SPM in Jan** | | | | | | | | | | | | | |
| Water-1 m | 112° 14'28.453" | 38° 54'33.980" | | 1.00 | 0.16 | 2.88 | 0.25 | 0.27 | -0.75 | -6.32 | 6.85 | 10.40 | 7.95 |
| Water-3 m | 112° 14'28.453" | 38° 54'33.980" | | 3.00 | 0.36 | 6.09 | 0.26 | 0.26 | -0.49 | -5.57 | 7.12 | 11.02 | 7.77 |
| Water-6 m | 112° 14'28.453" | 38° 54'33.980" | | 6.00 | 0.49 | 8.05 | 0.25 | 0.27 | -0.65 | -6.24 | 6.95 | 10.57 | 7.99 |
| Water-8 m | 112° 14'28.453" | 38° 54'33.980" | | 8.00 | 0.22 | 3.71 | 0.24 | 0.28 | -0.96 | -6.89 | 6.63 | 10.20 | 8.24 |

MAAT represents mean annual air temperature.

[a] Calculated according to Eq. (9).

[b] Calculated according to Eq. (10).

[c] and [d] Calculated according to Eq. (16) and (17).

[e] Calculated according to Eq. (15).

**Table 2** Calibrations for brGDGT-derived temperature proxies used in this study.

| Calibrations | Equation no. in the text | References |
|---|---|---|
| **For soils** | | |
| MAAT=0.81-5.67*CBT+31.0*MBT' ($n$=176, $r^2$=0.59, RMSE=5.0 ˚C) | (8) | Peterse et al. (2012) |
| MAAT=-8.57+31.45*MBT'$_{5ME}$ ($n$=222, $r^2$=0.66, RMSE=4.8 ˚C) | (9) | De Jonge et al. (2014) |
| MAAT [a]=27.63*Index 1-5.72 ($n$=148, r2=0.75, RMSE=2.5 ˚C) | (10) | Wang et al. (2016) |
| **For sediments** | | |
| MAAT=6.803-7.062*CBT+37.09*MBT ($n$=139, $r^2$=0.62, RMSE=5.24 ˚C) | (11) | Global, Sun et al. (2011) |
| MAAT=8.263-17.938*CBT+46.675*MBT ($n$=24, $r^2$=0.52, RMSE=5.1 ˚C) | (12) | Regional, Sun et al. (2011) |
| MAAT [b]=50.47-74.18*f(IIIa)-31.60*f(IIa)-34.69*f(Ia) ($n$=46, $r^2$=0.94, RMSE=2.2 ˚C) | (13) | Tierney et al. (2010) |
| MAAT=22.77-33.58*f(IIIa)-12.88*f(IIa)-418.53*f(IIc)+86.43*f(Ib) ($n$=111, $r^2$=0.94, RMSE=1.9 ˚C) | (14) | Loomis et al. (2012) |
| Growth AT=21.39*MBT'$_{6ME}$+2.27 ($n$=39, $r^2$=0.75, RMSE=1.78 ˚C) | (15) | Dang et al. (2018) |
| MAAT=23.81-31.02*f(IIIa)-41.91*f(IIb)-51.59*f(IIb')-24.70*f(IIa)+68.80*f(Ib) ($n$=65, $r^2$=0.94, RMSE=2.14 ˚C) | (16) | Russell et al. (2018) |
| MAAT=-1.21+32.42*MBT'$_{5ME}$ | (17) | Russell et al. (2018) |

AT represents air temperature.

MAAT represents mean annual air temperature.

[a] Index=log[(Ia+Ib+Ic+IIa'+IIIa')/(Ic+IIa+IIc+IIIa+IIIa')].

[b] Fractional abundance of brGDGTs is a fraction of only brGDGT Ia, IIa and IIIa.

**Table 3** Comparison of measured air temperature, brGDGT-derived temperature from catchment soils and

 brGDGT-derived temperature from sediments in different lake basins.

| Name | Latitude | Longitude | Depth (m) | MAAT (°C) | Mean warm season AT (°C) | Mean annual LWT (°C) | Surface soils MAAT[a] (°C) | Surface sediments | | | | References |
|---|---|---|---|---|---|---|---|---|---|---|---|---|
| | | | | | | | | MAAT[b] (°C) | MAAT[c] (°C) | MAAT[d] (°C) | MAAT[e] (°C) | |
| Gonghai Lake | 38°54′N | 112°14′E | 9 | 4.3 | 12.1 | 10.6 | 3.96±1.46 | 10.74±0.33 | 9.70±0.71 | 10.86±1.33 | 7.93±1.46 | Cao et al. (2017) |
| Lake Towuti | 2.5°S | 121°E | 200 | 24 | 24 | n.d. | 22.52±2.61 | 26.62±1.10 | 29.13±1.86 | n.d. | n.d. | Tierney and Russell (2009) |
| Lake Huguanyan | 21°09′N | 110°17′E | 20 | 23.2 | 23.2 | 24.8 | 23.80±1.39 | 25.11±0.60 | 28.12±0.90 | 26.47±0.83 | 26.07±0.73 | Hu et al. (2015, 2016) |
| Lake Donghu | 30°54′N | 114°41′E | 6 | 16 | 16 | 20 | 15.79±4.37 | 19.74±0.39 | 22.82±0.51 | 25.75±0.34 | 20.61±0.71 | Qian et al. (2019) |
| Qinghai Lake | 36°54′N | 100°01′E | 27 | 0.65 | 7 | n.d. | 3.38±2.40 | 12.34±0.87 | 9.92±1.14 | 13.61±1.49 | 8.80±1.11 | Wang et al. (2012) |
| Lower King pond | 44°25′N | 72°26′W | 8 | 6 | 11.3 | 11.6 | 11.50±2.08 | 14.97±0.42 | 14.9±0.53 | 18.75±0.64 | 15.76±0.84 | Loomis et al. (2014) |

AT represents air temperature and MAAT represents mean annual air temperature.

LWT represents lake water temperature.

[a] Calculated after according to Eq. (8).

[b] and [c] Calculated according to Eq. (11) and (12).

 [d] Calculated after according to Eq. (13).

[e] Calculated after according to Eq. (14).