# Peer review of "Ice formation on lake surface in winter causes warm season bias of lacustrine brGDGT temperature estimates"

_Biogeosciences, 2019_

## Referee Comment (RC1) · Anonymous Referee #1 · 19 Feb 2020

The authors of this manuscript examine the brGDGT distributions in the water column and surface sediments of the Lake Gonghai and its catchment. They address a critical issue for brGDGT studies which is the warm season bias of brGDGT-derived temperatures obtained in lakes. They propose a new very interesting mechanism to explain this bias implying the decoupling of air and lake water temperature during the cold season due to ice formation. This finding will be useful for the community and is worthy to be published. However, some improvements can be made before publication.

Main comments 1. The new separation method of the 5 and 6-Me isomers should be mentioned in the introduction. 2. A figure with the different forms of brGDGTs could be

included in appendix. 3. Why don't you use recent regional soil calibrations for China as the one of Wang et al., 2016 for your soil samples? 4. The conclusion is incomplete, you could add that soil temperature reconstructions reflect the MAAT and I think that it is important to mention that brGDGT distributions in the water column change with seasons while brGDGT productivity does not seem to significantly change. This allows you to propose an alternative explanation to warm season bias in brGDGT-derived temperature that is currently mainly considered as linked with changing brGDGT productivity. 5. The manuscript should be carefully checked for grammar and language issues.

Abstract

117 mean annual

I18 There are too many 'and'.

I29 I think that the use of 'believe' should be avoided and the sentence should be rewritten. Suggestion: we think that lacustrine brGDGTs actually reflect the mean annual LWT (...).

Introduction

I42 The abbreviations MBT and CBT should be defined.

146 Some references could be added in particular, recent ones using the new separation method.

I53 Suggestion: brGDGTs could be produced in situ in lake environments and differ significantly from soil derived brGDGTs (...)

1106 'composition distribution of brGDGTs' sounds odd to me, I suggest you to change it in the entire manuscript and replace it by 'brGDGT distribution'.

I107 and further discuss ...

Materials and methods

I112 Mention 'N' and 'E' for latitude and longitude.

1119 concentrated

I146 combination of

1146 and 150 'DCM' and 'MeOH' could be used for dichloromethane and methanol defining the acronym at the first appearance.

I157 Mention what are 'A' and 'B'.

I167 Remove the ';' after(2)

1169 The authors could mention Martin et al., 2019 who modified the initial definition of the IIIa/IIa ratio proposed by Xiao et al., 2016.

1170 A word is missing as well as a punctuation mark.

1177 Add a figure in appendix describing the different brGDGT structures and refer to it here.

Results

I206 typical for in situ produced lacustrine (...)

Discussion

I221 This title is not very clear, maybe 'Different sources of brGDGTs in the Gonghai Lake' or 'In situ production of brGDGTs in the Gonghai Lake'.

I226 and/or surface sediments. I would not mention brGDGT concentrations as a discriminant factor between soils and in situ production, differences of concentrations, alone, would not be a proof of the occurrence of in situ production as several other parameters could be involved.

I227 comparison of brGDGT distribution

I232 was similar to that of SPM ... from that of soils

СЗ

I236 The  $\Sigma$ IIIa/ $\Sigma$ IIa values in sediments and SPM were

I238 the  $\Sigma IIIa/\Sigma IIa$  ratio in sediments and SPM was significantly higher than in catchment soils.

l239 sediments are

I243 It does not appear very clearly that #Ringstetra were higher in sediments than in soils, a statistical test would be appreciated.

I257-258 in globally distributed lakes?

I270 You should provide the reader with the analytical error associated with the MBT indices in the method section for a better evaluation of the changes discussed here.

I274 You should add a reference to Fig. 2. You should at least mention that the deepest SPM shows an opposite trend which seems to indicate that at this depth, temperature is not the only parameter controlling brGDGT distributions.

I275 seasonal temperature changes?

I276 previously

I276-277 The phrasing sounds odd to me. Suggestion: suggest that both ... could respond to temperature changes.

I278-279 African ; the phrasing is not very clear here.

I283 I suppose that 0.3 corresponds to the difference of the mean temperatures obtained for September and July? You could specify it.

l284 remove the '.' after (16).

l286 multi-seasonal

l287 previously

l292 could also be

1293 the shallow water depth of the lake

I293-294 The sentence is not very clear and too long, you should maybe cut it into two sentences.

I296-299 Terrestrial inputs are almost not discussed, could they have a role in seasonal changes of brGDGTs?

I297 Be coherent with the notation of  $\Sigma$ IIIa/ $\Sigma$ IIa.

l300-304 You should mention here or in the previous paragraphs that SPM samples also reflect temperatures close to warm season AT.

I302 You could add a reference to the table 1. Is it 13.2 or 13.5?

I305 correlated significantly

I314 are thought to ...

l314-325 You say in situ production is thought to be the main source of brGDGTs in many lakes so why do you only consider six lakes in fig 5? What about the others?

I319-325 Rephrase

I332 You should also mention that brGDGT distribution in water column seems to change seasonally in agreement with temperature, what you discussed in the 4.3 section.

I339 Rephrase

I350 Most of stratified lakes ...

l365 Can you add a reference?

1378 universal

I383 compatible with the mechanism that we propose here

I391 Use cold season rather than 'winter'.

Conclusion

I403 from that in soils

I403-404 indicating that lacustrine brGDGTs are mainly produced in situ ...

1404 in surface sediments of Lake Gonghai

1406 water-air decoupling in Lake Gonghai

Figure 1

(a) northern limit ? (b) For the latitude replace 'E' by N I655 represent

Figure 3

fractional abundance I666 Maybe specify: water column (SPM). I667 Use degree of cyclisation rather than 'cyclisation ratio'.

Figure 4

l670 soil calibration.

Figure 5

I679 A comma missing before Lake Donghu. I676 Mention the number of the equation used.

Table 1

I695 For b et c, mention the number of the equation used.

Table 3

1705-708 Mention the number of the equation used.

---

## Referee Comment (RC2) · Anonymous Referee #2 · 2 Mar 2020

General comments The mechanism of season bias of lake brGDGTs-derived temperature is not very clear, hence limit the application of brGDGTs index in lakes. The manuscript proposes a new idea about this hot topic. They conclude that decoupling of water and air temperature in winter causes warm season bias of lacustrine brGDGTs temperature estimates. Therefore, I recommend this manuscript for publication in the journal after improvement.

Detailed comments 1/ Many pervious researchers suggested that soil calibrations could not be applicable to lake sediments for temperature reconstruction, if aquatic production of brGDGTs is predominant over soil input (e.g. many papers). It is no new,
and not necessary to discuss too much in this point in your manuscript. And to focus on SPM.

2/ Seasonality is a major feature for almost all organic proxies . For example, Lake Huguangyan (Hu et al.,2016; Chu et al., 2017). Lake limnology is most important, for example, Lake Huguangyan is a monomictic lake.

3/ "Line 147-148: " There is no water column stratification whether summer or winter". You must revise this sentence. Based on the location and depth of the lake, it might be stratified in summer. And figure 2 shows a little stratification occurred in September(autumn).

4/ Line 360-365: I don't think the estimated temperature using the calibration of Dang et al. (2018) are close to the mean warm season AT in GH, even if the RMSE is being considered. It seems that the calibration of Ressell et al. (2018) may be more suitable for your explanation, and you'd get more discuss about this point.

5/ Line 450: The definition of warm season should be given earlier, and change "monthly temperature" to "average monthly temperature".

6/ Line 464-465: "For example, 464 MBT/CBT-derived temperature correlated better with warm season AT than with annual mean AT in the tropical Lake Huguangyan, suggesting a warm season bias (Sun et al., 2011)". To improve the discussion of seasonality in the paper, I recommend authors should detailed read the paper of Sun et al. (2011) carefully. And the author should see discussion about the seasonality of brGDGTs in Lake Huguangyan from Hu et al. (2016) and Chu et al. (2017). Seasonal biases may be due to seasonal brGDGTs production, and link to lake limnology and local climate.

7/ Please provide the component specific content of brGDGT as a Supplement.

8/ This manuscript is worth publish because something is new. But, authors should mention that the limited data in your manuscript, and more works are need to verify

this question.

---

## Author Comment (AC1) · 13 Mar 2020

Response to "Interactive comment on 'Decoupling of water and air temperature in winter causes warm season bias of lacustrine brGDGTs temperature estimates'by Anonymous Referee #1"

The authors of this manuscript examine the brGDGT distributions in the water column and surface sediments of the Lake Gonghai and its catchment. They address a critical issue for brGDGT studies which is the warm season bias of brGDGT-derived temperatures obtained in lakes. They propose a new very interesting mechanism to explain this bias implying the decoupling of air and lake water temperature during the cold season

due to ice formation. This finding will be useful for the community and is worthy to be published. However, some improvements can be made before publication.

Response: Thanks for the comments. We have made substantial improvements according to reviewers' suggestions. Besides, some improvements were made beyond those suggestions during our revision, including title rephrasing, reanalysis sedimentary dada instead of presenting mean brGDGT values (line 1, 22-24, 216-217, 224-225, 241-242, 244-246), and reorganization of discussion (line 258-259, 382-418, 426-433, 447-471). We think the manuscript has been greatly improved in logic.

Main comments

1. The new separation method of the 5 and 6-Me isomers should be mentioned in the introduction.

Response: We agree and have added related contents about 5 and 6-Me isomers in the introduction. Please see line 48-55 in the revision. The revised sentences are following: 'With improved analytical methods, a series of 6-methyl brGDGTs, previously co-eluted with 5-methyl brGDGTs, were identified (De Jonge et al., 2013), which may introduce scatter in the original MBT'/CBT calibration for the mean annual AT (De Jonge et al., 2014). Thus, exclusion of the 6-methyl brGDGTs from the MBT', i.e. the newly defined MBT'5ME, results in improved calibrations (De Jonge et al., 2014; Wang et a., 2016; Wang et al., 2019). Calibrations using globally distributed surface soils for the MBT/CBT, MBT'/CBT or MBT'5ME indices (Weijers et al., 2007a; Peterse et al., 2012; De Jonge et al., 2014) have been widely used for continental AT reconstruction (e.g., Weijers et al., 2007b; Niemann et al., 2012; Lu et al., 2019)'.

2. A figure with the different forms of brGDGTs could be included in appendix.

Response: Done. Figure A1 showing different forms of brGDGTs has been added in Appendix 1.

3. Why don't you use recent regional soil calibrations for China as the one of Wang et

al., 2016 for your soil samples?

Response: Thanks for your suggestion. The regional soil calibration from Wang et al. (2016) has been applied to the Gonghai Lake, yielding $-2.42 \pm 1.19$ °C from soils, $-5.86 \pm 1.30$ °C from lake sediments (Table 1), $-6.20 \pm 0.60$ °C in Sept (Table 1) and $-6.25 \pm 0.54$ °C in Jan from SPM (Table 1). These values are significantly different from actual values, suggesting that the regional soil calibration was not suitable for soil and lake temperature reconstruction in the Gonghai Lake basin. We have added the results about brGDGT calculated temperature values from Wang et al. (2016) in the text, Table 1 and calibration in Table 2. Please see Line 279-284, Table 1 and 2 in the revision.

4. The conclusion is incomplete, you could add that soil temperature reconstructions reflect the MAAT and I think that it is important to mention that brGDGT distributions in the water column change with seasons while brGDGT productivity does not seem to significantly change. This allows you to propose an alternative explanation to warm season bias in brGDGT-derived temperature that is currently mainly considered as linked with changing brGDGT productivity.

Response: Thanks for your suggestion. The conclusion has been rewritten accordingly. Please see lines 511-529 in the revision.

5. The manuscript should be carefully checked for grammar and language issues.

Response: Done, thanks for your suggestion.

Abstract

l17 mean annual

Response: Done. Please see line 17 in the revision.

l18 There are too many 'and'.

Response: This sentence has been rephrased to 'we investigated the brGDGTs from

catchment soils, suspended particulate matter (SPM) and surface sediments in the Gonghai Lake in north China to explore this question'. Please see line 19 in the revision.

l29 I think that the use of 'believe' should be avoided and the sentence should be rewritten. Suggestion: we think that lacustrine brGDGTs actually reflect the mean annual LWT (. . .).

Response: Done as you suggested. Please see line 32 in the revision.

Introduction

l42 The abbreviations MBT and CBT should be defined.

Response: Done. Please see line 45-46 in the revision.

l46 Some references could be added in particular, recent ones using the new separation method.

Response: Done. References 'De Jonge et al. (2014), Wang et al. (2016), Wang et al. (2019)' have been added in line 52.

l53 Suggestion: brGDGTs could be produced in situ in lake environments and differ significantly from soil derived brGDGTs (. . .)

Response: Done as you suggested. Please see line 63 in the revision.

l106 'composition distribution of brGDGTs' sounds odd to me, I suggest you to change it in the entire manuscript and replace it by 'brGDGT distribution'.

Response: Done. Please see line 63, 69, 116-117, 254, 260, 511, 514, 796 in the revision.

l107 and further discuss . . .

Response: Done. Please see line 118 in the revision.

Materials and methods

l112 Mention 'N' and 'E' for latitude and longitude.

Response: Done. Please see line 123 in the revision.

l119 concentrated

Response: Changed. Please see line 130 in the revision.

l146 combination of

Response: Changed. Please see line 161 in the revision.

l146 and 150 'DCM' and 'MeOH' could be used for dichloromethane and methanol defining the acronym at the first appearance.

Response: Done. Please see line 160 and 165 in the revision.

l157 Mention what are 'A' and 'B'.

Response: Done. Please see line 172 in the revision.

l167 Remove the ';' after(2)

Response: Done. Please see line 183 in the revision.

l169 The authors could mention Martin et al., 2019 who modified the initial definition of the IIIa/IIa ratio proposed by Xiao et al., 2016.

Response: Done. Please see line 184-185 in the revision.

l170 A word is missing as well as a punctuation mark.

Response: Done. Please see line 186 in the revision.

l177 Add a figure in appendix describing the different brGDGT structures and refer to it here.

Response: Done. We have added the figure about brGDGT structures in Appendix 1.

Results

l206 typical for in situ produced lacustrine (. . .)

Response: Done. Please see line 221 in the revision.

Discussion

l221 This title is not very clear, maybe 'Different sources of brGDGTs in the Gonghai Lake' or 'In situ production of brGDGTs in the Gonghai Lake'.

Response: Thanks for your suggestion. We have changed 'Different sources of lacustrine brGDGTs from surrounding soils' to 'In situ production of brGDGTs in the Gonghai Lake'. Please see line 249 in the revision.

l226 and/or surface sediments. I would not mention brGDGT concentrations as a discriminant factor between soils and in situ production, differences of concentrations, alone, would not be a proof of the occurrence of in situ production as several other parameters could be involved.

Response: We found content of brGDGTs in surface sediments is significantly higher than that in surface soils (Table 1), and increases with water depth (Table 1). Therefore, we think it suggests a possible autochthonous contribution in Gonghai Lake. We have added 'Moreover, they exhibited a clearly increasing trend with water depth' in the revision, please see line 258-259.

l227 comparison of brGDGT distribution

Response: Changed. Please see line 260 in the revision.

l232 was similar to that of SPM . . . from that of soils

Response: Changed. Please see line 260-261 in the revision.

l236 The $\Sigma$IIIa/$\Sigma$IIa values in sediments and SPM were

Response: Changed. Please see line 266 in the revision.

l238 the $\Sigma$IIIa/$\Sigma$IIa ratio in sediments and SPM was significantly higher than in catchment soils.

Response: Changed. Please see line 266 in the revision.

l239 sediments are

Response: Changed. Please see line 269 in the revision.

l243 It does not appear very clearly that #Ringstetra were higher in sediments than in soils, a statistical test would be appreciated.

Response: Thanks for your suggestion. We have added t-test in the revised sentence, please see '... #Ringstetra and #Ringspenta 5ME were clearly higher in sediments than in catchment soils (p<0.05 for #Ringstetra, p<0.01 for #Ringspenta 5ME), although #Ringspenta 6ME in sediments was similar to that in catchment soils (p=0.11 for #Ringspenta 6ME; Fig. 3b)' in line 273-275.

l257-258 in globally distributed lakes?

Response: We have changed 'in global lakes' to 'in many modern lake sediments'. Please see line 288 in the revision.

l270 You should provide the reader with the analytical error associated with the MBT indices in the method section for a better evaluation of the changes discussed here.

Response: We have added the analytical error in the method. Please see 'Based on duplicate HPLC/MS analyses, the analytical errors of both the MBT'5ME and MBT'6ME index were ±0.01 units' in line 177-188 in the revision.

l274 You should add a reference to Fig. 2. You should at least mention that the deepest SPM shows an opposite trend which seems to indicate that at this depth, temperature is not the only parameter controlling brGDGT distributions.

Response: In revised Fig. 2, the MBT'5ME and MBT'6ME index trace the water temperature changes at different depth in Sept and in Jan, and it seems that MBT'5ME and MBT'6ME index could response to water temperature changes to some extent. However, the seasonal changes of SPM brGDGT derived temperature between Sept and Jan were small, which could be influenced by the several reasons in addition to water temperature, such as residence of "fossil" brGDGTs and sediment resuspension, as evidence of smaller differences in MBT'5ME and MBT'6ME between sediments and SPM at deeper sites. We have discussed the detailed reason about it in the later paragraph. Please see line 391-409 in the revision.

l275 seasonal temperature changes?

Response: This paragraph have been rephrased. Please see line 383-390 in the revision.

l276 previously Response: This paragraph have been rephrased. Please see line 383-390 in the revision.

l276-277 The phrasing sounds odd to me. Suggestion: suggest that both . . . could respond to temperature changes.

Response: This paragraph have been rephrased. Please see line 383-390 in the revision.

l278-279 African ; the phrasing is not very clear here.

Response: This paragraph have been rephrased. Please see line 383-390 in the revision.

l283 I suppose that 0.3 corresponds to the difference of the mean temperatures obtained for September and July? You could specify it.

Response: We have revised sentence as 'Although the MBT'5ME and MBT'6ME in SPM in the lake seem to reflect temperature changes in the water column to some

extent, the differences of brGDGT-derived temperatures based on lake-specific calibrations between September and January ($-0.93$–$1.21$ °C) are much lower than the measured difference ($\sim$13 °C), independent of the calibration of (15), (16) or (17) (Tables 1 and 2)'. This could be clear for reader. Please see line 391-394 in the revision.

l284 remove the '.' after (16).

Response: Done. Please see line 394 in the revision.

l286 multi-seasonal

Response: Changed. Please see line 401 in the revision.

l287 previously

Response: This paragraph have been rephrased. Please see line 391-409 in the revision.

l292 could also be

Response: This paragraph have been rephrased. Please see line 391-409 in the revision.

l293 the shallow water depth of the lake

Response: This paragraph have been rephrased. Please see line 391-409 in the revision.

l293-294 The sentence is not very clear and too long, you should maybe cut it into two sentences.

Response: This paragraph have been rephrased. Please see line 391-409 in the revision.

l296-299 Terrestrial inputs are almost not discussed, could they have a role in seasonal changes of brGDGTs?

Response: Just as discussed in the text, several parameters, such as $\Sigma$IIIa/$\Sigma$IIa, IR6ME, #Ringstetra and #Ringspenta in SPM were in-between the soil and sediment values, we speculate terrestrial inputs may be a factor, if any, to reduce the seasonal changes of brGDGTs in SPM. Please see line 407-409 in the revision.

l297 Be coherent with the notation of $\Sigma$IIIa/$\Sigma$IIa.

Response: Done. Please see line 407 in the revision.

l300-304 You should mention here or in the previous paragraphs that SPM samples also reflect temperatures close to warm season AT.

Response: Done. Sedimentary brGDGTs in Gonghai Lake reflected temperature close to warm season AT. Due to sediment resuspension, the warm season bias also occurred in SPM. We have discussed it in line 428-433 in the revision.

l302 You could add a reference to the table 1. Is it 13.2 or 13.5?

Response: Done. The number of 13.5 is correct according to the results in Table 1.

l305 correlated significantly

Response: Done. This paragraph have been rephrased and this word has been deleted.

l314 are thought to . . .

Response: Done. This sentence has been rephrased. Please line 356-358 in the revision.

l314-325 You say in situ production is thought to be the main source of brGDGTs in many lakes so why do you only consider six lakes in fig 5? What about the others?

Response: We selected these five lakes for several reasons. (i) The brGDGTs-related data from both the lake surface sediments and the catchment surface soils were available. (ii) The lakes are in different regions, and together with Gonghai Lake in this

study, they span a relatively large environmental gradient. (iii) The authors of these studied lakes have claimed that brGDGT distribution in lake sediments differed from catchment soils as we do in Gonghai Lake in this study. As to others lakes, due to the lack of catchment soil brGDGT data, they are not shown in revised Figure 5, although brGDGT-derived temperatures are also warm season biased. We have added related content in the revised manuscript. Please see line 358-381 in the revision.

l319-325 Rephrase

Response: Done. Please see line 371-381 in the revision.

l332 You should also mention that brGDGT distribution in water column seems to change seasonally in agreement with temperature, what you discussed in the 4.3 section.

Response: Although the MBT'5ME and MBT'6ME in water SPM mirror the water temperature change in Sept and Jan, the calculated seasonal temperature offsets was quite small in Gonghai Lake. So we don't emphasize this phenomenon. We have added related content in the revised manuscript. Please see line 391-409 in the revision.

l339 Rephrase

Response: Done. Please see line 441-444 in the revision.

l350 Most of stratified lakes . . .

Response: Done. This paragraph has been rephrased and this word has been deleted.

l365 Can you add a reference?

Response: Done. Please see line 474 in the revision.

l378 universal

Response: Changed. Please see line 487 in the revision.

[Figure]

l383 compatible with the mechanism that we propose here

Response: Changed. Please see line 492 in the revision.

l391 Use cold season rather than 'winter'.

Response: This paragraph has been deleted.

Conclusion

l403 from that in soils

Response: Changed. Please see line 515 in the revision.

l403-404 indicating that lacustrine brGDGTs are mainly produced in situ . . .

Response: Changed. Please see line 515 in the revision.

l404 in surface sediments of Lake Gonghai

Response: Changed. Please see line 521 in the revision.

l406 water-air decoupling in Lake Gonghai

Response: We don't add "in Lake Gonghai" here because we think the sentence is ok.

Figure 1

(a) northern limit ? (b) For the latitude replace 'E' by N l655 represent

Response: Changed. Please see in the revised Figure 1.

Figure 3

fractional abundance Maybe specify: water column (SPM).

Response: Changed. Please see line 796 in the revision.

l667 Use degree of cyclisation rather than 'cyclisation ratio'.

Response: Done. Please see line 797-798 in the revision.

Figure 4

l670 soil calibration.

Response: This figure have been replace, please see revised caption Figure 4.

Figure 5

l679 A comma missing before Lake Donghu.

Response: Changed. Please see line 811 in the revision.

l676 Mention the number of the equation used.

Response: Done. Please see line 807-810 in the revision.

Table 1

l695 For b et c, mention the number of the equation used.

Response: Done. Please see the note of Table 1 in the revision.

Table 3

l705-708 Mention the number of the equation used.

Response: Done. Please see the note of Table 3 in the revision.

Please also note the supplement to this comment:
https://www.biogeosciences-discuss.net/bg-2019-507/bg-2019-507-AC1-supplement.pdf

———————————————————————

**Fig. 1.**

[Figure]

**Fig. 2.**

[Figure]

[Figure]

**Fig. 3.**

[Figure]

[Figure]

**Fig. 4.**

[Figure]

[Figure]

[Figure]

**Fig. 5.**

**Supplement:**

[Figure]

**Appendix 1** Molecular structures of brGDGTs

---

## Author Response (AR1)

**Associate Editor comment:**

First of all, I would like to thank both reviewers for their reviews and you for your reply. I have read you manuscript with great pleasure, but agree with the comments of the reviewers. So please do address these in you next version of the manuscript.

Response: Thank for your handling our manuscript. We found your comments below are very constructive for improving our manuscript. The reviewers' and your comments have been address in the updated version.

I do have a few other questions or suggestions. I totally agree that if the majority of your brGDGTs are produced within the lake, lake water temperature is probably more important than air temperature. That being said, there is still the option that soil derived brGDGTs create part of the observed bias, right? In soils I would assume production is highest during the warm season and decreases or even completely stops in the cold season, when the soils are probably also frozen? What is the catchment of this lake and when does soil derived material end up in the lake? I could imagine soil brGDGTs with a warm season bias being stored in soils over winter ending up in the lake with the melting of snow and ice transporting soil derived material with meltwater to the lake ending up in the sediment. If something like this might happen, you would probably see such as soil signal in mainly the spring SPM and maybe only in certain areas of the lake. Did you sample in spring? So big question is the soil contribution always small and insignificant relative to the lake contribution? Or might there be a small, but significant, contribution to the sediment creating this warm bias? I totally agree with your lake temperatures and how they deviate from the air temperatures in winter, so a very valid explanation, but can you completely rule out the soil contribution based on the data set you have presented here?

Response: (1) We admit that we cannot estimate soil contribution to lake sediment due to lack of any

suitable index to completely separate the two brGDGT pools at present. So what we do in our work is to compare brGDGT contents and distributions between sediment and soils, and qualitatively say that soil contribution is quite minor to sediments (please see Line 250-290 in the revision). (2) The Gonghai Lake is a closed alpine lake without river input and output. The catchment soil input to the lake occurs mainly from May to September (the warmest months), when ca. 80% annual rainfall and enhanced erosion take place. Spring is not an important season for soil input. (3) We have been also interested in the assumption of highest brGDGT production during the warm season in catchment soils, which, as you said, may finally create part of the observed bias. However, our data do not support this idea, because the brGDGT-derived temperature in the Gonghai catchment soils using global and regional calibrations are close to or even lower than the mean annual temperature (please see Line 278-284 in the revision).

That was one potential issue. What is reflected in you SPM, is there the potential for resuspension, for instance? I would expect activity and/or growth also to be reduced at 5 °C relative to the 14, 15, 20 °C in summer. Agreed potentially not zero, which would most likely be the case for the frozen soils. Your lake temperature lag behind the air temperature a little, the brGDGTs possibly integrate a relatively long(er) time period, even in the SPM, and therefore lag behind even more or average out a longer time period. Add to that the possibility of soil derived material which could be seasonally varying and possibly resuspension. Could it be that the relatively stable amount of brGDGTs in SPM reflects these process, slow growth and a "fossil" component leading to averaging out of the extremes and resulting in an apparent warm bias?

Response: (1) You are right. We have elaborated the "fossil" nature of brGDGTs in our SPM samples (Please see Line 386-409 in the revision). (2) Your comment of slow growth of bacteria under low temperatures and hence causing warm bias of brGDGT distribution is suggestive. However, in our

results brGDGT-derived temperatures in SPM were close to mean annual water temperature and lower than the mean annual warm water temperature. So our data do not support this idea. We add this point in our text (Please see Line 446-449 in the revision).

There one last remark, would it be possible that the population growing and active at 5 °C does something different than the one growing in summer in the lake? Something like this is happening in soils where the observed differences are related to population changes and not adaptations to changing conditions by the same population. If so, how would that interfere with your ideas?

Response: This is a clever comment. We admit we cannot give a perfect answer to this question due to lack of supported data. Nonetheless, we tried to make a simple discussion on this idea. Please see Line 419-433 in the revision.

Again, I think your idea is very valid, but I do think some of these other complicating mechanisms could be discussed. I assume you did not analyse a spring sample, or a sample from during or right after a major ice and snow melt. 14C age data of brGDGTs from different samples would also be very interesting, I think, but that is a completely different topic, never mind.

Response: Thanks for your constructive comments that help improve our manuscript greatly.

**Anonymous Referee #1**

Received and published: 19 February 2020

The authors of this manuscript examine the brGDGT distributions in the water column and surface sediments of the Lake Gonghai and its catchment. They address a critical issue for brGDGT studies which is the warm season bias of brGDGT-derived temperatures obtained in lakes. They propose a new very interesting mechanism to explain this bias implying the decoupling of air and lake water temperature during the cold season due to ice formation. This finding will be useful for the community and is worthy to be published. However, some improvements can be made before publication.

Response: Thanks for the comments. We have made substantial improvements according to reviewers' suggestions. Besides, some improvements were made beyond those suggestions during our revision, including title rephrasing, reanalysis sedimentary data instead of presenting mean brGDGT values (line 1, 22-24, 216-217, 224-225, 241-242, 244-246), and reorganization of discussion (line 258-259, 382-418, 426-433, 441-483). We think the manuscript has been greatly improved in logic.

Main comments:

1. The new separation method of the 5 and 6-Me isomers should be mentioned in the introduction.

Response: We agree and have added related contents about 5 and 6-Me isomers in the introduction. Please see line 48-55 in the revision. The revised sentences are following:

'With improved analytical methods, a series of 6-methyl brGDGTs, previously co-eluted with 5-methyl brGDGTs, were identified (De Jonge et al., 2013), which may introduce scatter in the original MBT'/CBT calibration for the mean annual AT (De Jonge et al., 2014). Thus, exclusion of the 6-methyl brGDGTs from the MBT', i.e. the newly defined MBT'5ME, results in improved calibrations (De Jonge et al., 2014; Wang et a., 2016; Wang et al., 2019). Calibrations using globally distributed surface soils for the MBT/CBT, MBT'/CBT or MBT'5ME indices (Weijers et al., 2007a; Peterse et al., 2012; De Jonge et al., 2012; Lu et al., 2019)'.

2. A figure with the different forms of brGDGTs could be included in appendix.

Response: Done. Figure A1 showing different forms of brGDGTs has been added in Appendix 1.

3. Why don't you use recent regional soil calibrations for China as the one of Wang et al., 2016 for your soil samples?

Response: Thanks for your suggestion. The regional soil calibration from Wang et al. (2016) has been applied to the Gonghai Lake, yielding  $-2.42 \pm 1.19$  °C from soils,  $-5.86 \pm 1.30$  °C from lake sediments (Table 1),  $-6.20 \pm 0.60$  °C in Sept (Table 1) and  $-6.25\pm0.54$  °C in Jan from SPM (Table 1). These values are significantly different from actual values, suggesting that the regional soil calibration was not suitable for soil and lake temperature reconstruction in the Gonghai Lake basin.

We have added the results about brGDGT calculated temperature values from Wang et al. (2016) in the text, Table 1 and calibration in Table 2. Please see Line 279-284, Table 1 and 2 in the revision.

4. The conclusion is incomplete, you could add that soil temperature reconstructions reflect the MAAT and I think that it is important to mention that brGDGT distributions in the water column change with seasons while brGDGT productivity does not seem to significantly change. This allows you to propose an alternative explanation to warm season bias in brGDGT-derived temperature that is currently mainly considered as linked with changing brGDGT productivity.

Response: Thanks for your suggestion. The conclusion has been rewritten accordingly. Please see lines 524-542 in the revision.

5. The manuscript should be carefully checked for grammar and language issues. Response: Done, thanks for your suggestion.

Abstract

117 mean annual

Response: Done. Please see line 17 in the revision.

118 There are too many 'and'.

Response: This sentence has been rephrased to 'we investigated the brGDGTs from catchment soils, suspended particulate matter (SPM) and surface sediments in the Gonghai Lake in north China to explore this question'. Please see line 19 in the revision.

129 I think that the use of 'believe' should be avoided and the sentence should be rewritten. Suggestion: we think that lacustrine brGDGTs actually reflect the mean annual LWT (...).

Response: Done as you suggested. Please see line 32 in the revision.

Introduction

142 The abbreviations MBT and CBT should be defined.

Response: Done. Please see line 45-46 in the revision.

146 Some references could be added in particular, recent ones using the new separation method.

Response: Done. References 'De Jonge et al. (2014), Wang et al. (2016), Wang et al. (2019)' have been added in line 52.

153 Suggestion: brGDGTs could be produced in situ in lake environments and differ significantly from soil derived brGDGTs (...)

Response: Done as you suggested. Please see line 63 in the revision.

1106 'composition distribution of brGDGTs' sounds odd to me, I suggest you to change it in the entire manuscript and replace it by 'brGDGT distribution'.

Response: Done. Please see line 63, 69, 116-117, 254, 260, 524, 527, 812 in the revision.

1107 and further discuss . . .

Response: Done. Please see line 118 in the revision.

Materials and methods

1112 Mention 'N' and 'E' for latitude and longitude.

Response: Done. Please see line 123 in the revision.

1119 concentrated

Response: Changed. Please see line 130 in the revision.

1146 combination of

Response: Changed. Please see line 161 in the revision.

1146 and 150 'DCM' and 'MeOH' could be used for dichloromethane and methanol

defining the acronym at the first appearance.

Response: Done. Please see line 160 and 165 in the revision.

1157 Mention what are 'A' and 'B'.

Response: Done. Please see line 172 in the revision.

1167 Remove the ';' after(2)

Response: Done. Please see line 183 in the revision.

1169 The authors could mention Martin et al., 2019 who modified the initial definition of the IIIa/IIa

ratio proposed by Xiao et al., 2016.

Response: Done. Please see line 184-185 in the revision.

1170 A word is missing as well as a punctuation mark.

Response: Done. Please see line 186 in the revision.

1177 Add a figure in appendix describing the different brGDGT structures and refer to

it here.

Response: Done. We have added the figure about brGDGT structures in Appendix 1.

Results

1206 typical for in situ produced lacustrine (...)

Response: Done. Please see line 221 in the revision.

Discussion

1221 This title is not very clear, maybe 'Different sources of brGDGTs in the Gonghai Lake' or 'In situ production of brGDGTs in the Gonghai Lake'.

Response: Thanks for your suggestion. We have changed 'Different sources of lacustrine brGDGTs from surrounding soils' to 'In situ production of brGDGTs in the Gonghai Lake'. Please see line 249 in the revision.

1226 and/or surface sediments. I would not mention brGDGT concentrations as a discriminant factor between soils and in situ production, differences of concentrations, alone, would not be a proof of the occurrence of in situ production as several other parameters could be involved.

Response: We found content of brGDGTs in surface sediments is significantly higher than that in surface soils (Table 1), and increases with water depth (Table 1). Therefore, we think it suggests a possible autochthonous contribution in Gonghai Lake. We have added 'Moreover, they exhibited a clearly increasing trend with water depth' in the revision, please see line 258-259.

1227 comparison of brGDGT distribution

Response: Changed. Please see line 260 in the revision.

1232 was similar to that of SPM . . . from that of soils

Response: Changed. Please see line 260-261 in the revision.

1236 The  $\Sigma$ IIIa/ $\Sigma$ IIa values in sediments and SPM were

Response: Changed. Please see line 266 in the revision.

1238 the  $\Sigma$ IIIa/ $\Sigma$ IIa ratio in sediments and SPM was significantly higher than in catchment soils.

Response: Changed. Please see line 266 in the revision.

1239 sediments are

Response: Changed. Please see line 269 in the revision.

1243 It does not appear very clearly that #Ringstetra were higher in sediments than in soils, a statistical test would be appreciated.

8

Response: Thanks for your suggestion. We have added *t*-test in the revised sentence, please see '... #Ringstetra and #Ringspenta 5ME were clearly higher in sediments than in catchment soils (*p*<0.05 for #Ringstetra, *p*<0.01 for #Ringspenta 5ME), although #Ringspenta 6ME in sediments was similar to that in catchment soils (*p*=0.11 for #Ringspenta 6ME; Fig. 3b)' in line 273-275.

1257-258 in globally distributed lakes?

Response: We have changed 'in global lakes' to 'in many modern lake sediments'. Please see line 288 in the revision.

1270 You should provide the reader with the analytical error associated with the MBT indices in the method section for a better evaluation of the changes discussed here.

Response: We have added the analytical error in the method. Please see 'Based on duplicate HPLC/MS analyses, the analytical errors of both the MBT'5ME and MBT'6ME index were  $\pm 0.01$  units' in line 177-178 in the revision.

1274 You should add a reference to Fig. 2. You should at least mention that the deepest SPM shows an opposite trend which seems to indicate that at this depth, temperature is not the only parameter controlling brGDGT distributions.

Response: In revised Fig. 2, the MBT'5ME and MBT'6ME index trace the water temperature changes at different depth in Sept and in Jan, and it seems that MBT'5ME and MBT'6ME index could response to water temperature changes to some extent. However, the seasonal changes of SPM brGDGT derived temperature between Sept and Jan were small, which could be influenced by the several reasons in addition to water temperature, such as residence of "fossil" brGDGTs and sediment resuspension, as evidence of smaller differences in MBT'5ME and MBT'6ME between sediments and SPM at deeper sites. We have discussed the detailed reason about it in the later paragraph. Please see line 391-409 in the revision.

1275 seasonal temperature changes?

Response: This paragraph have been rephrased. Please see line 383-390 in the revision.

1276 previously

Response: This paragraph have been rephrased. Please see line 383-390 in the revision.

1276-277 The phrasing sounds odd to me. Suggestion: suggest that both . . . could respond to temperature changes.

Response: This paragraph have been rephrased. Please see line 383-390 in the revision.

1278-279 African ; the phrasing is not very clear here.

Response: This paragraph have been rephrased. Please see line 383-390 in the revision.

1283 I suppose that 0.3 corresponds to the difference of the mean temperatures obtained for September and July? You could specify it.

Response: We have revised sentence as

'Although the MBT'5ME and MBT'6ME in SPM in the lake seem to reflect temperature changes in the water column to some extent, the differences of brGDGT-derived temperatures based on lake-specific calibrations between September and January (-0.93–1.21 °C) are much lower than the measured difference (~13 °C), independent of the calibration of (15), (16) or (17) (Tables 1 and 2)'. This could be clear for reader. Please see line 391-394 in the revision.

1284 remove the '.' after (16).

Response: Done. Please see line 394 in the revision.

1286 multi-seasonal

Response: Changed. Please see line 401 in the revision.

1287 previously

Response: This paragraph have been rephrased. Please see line 391-409 in the revision.

1292 could also be

Response: This paragraph have been rephrased. Please see line 391-409 in the revision.

1293 the shallow water depth of the lake

Response: This paragraph have been rephrased. Please see line 391-409 in the revision.

1293-294 The sentence is not very clear and too long, you should maybe cut it into two sentences.

Response: This paragraph have been rephrased. Please see line 391-409 in the revision.

1296-299 Terrestrial inputs are almost not discussed, could they have a role in seasonal changes of brGDGTs?

Response: Just as discussed in the text, several parameters, such as  $\Sigma IIIa/\Sigma IIa$ , IR6ME, #Ringstetra and #Ringspenta in SPM were in-between the soil and sediment values, we speculate terrestrial inputs may be a factor, if any, to reduce the seasonal changes of brGDGTs in SPM. Please see line 407-409 in the revision.

1297 Be coherent with the notation of  $\Sigma IIIa/\Sigma IIa$ .

Response: Done. Please see line 407 in the revision.

1300-304 You should mention here or in the previous paragraphs that SPM samples also reflect temperatures close to warm season AT.

Response: Done. Sedimentary brGDGTs in Gonghai Lake reflected temperature close to warm season AT. Due to sediment resuspension, the warm season bias also occurred in SPM. We have discussed it in line 446-448 in the revision.

1302 You could add a reference to the table 1. Is it 13.2 or 13.5?

Response: Done. The number of 13.5 is correct according to the results in Table 1.

1305 correlated significantly

Response: Done. This paragraph have been rephrased and this word has been deleted.

1314 are thought to . . .

Response: Done. This sentence has been rephrased. Please line 356-358 in the revision.

1314-325 You say in situ production is thought to be the main source of brGDGTs in many lakes so

why do you only consider six lakes in fig 5? What about the others?

Response: We selected these five lakes for several reasons. (i) The brGDGTs-related data from both the lake surface sediments and the catchment surface soils were available. (ii) The lakes are in different regions, and together with Gonghai Lake in this study, they span a relatively large environmental gradient. (iii) The authors of these studied lakes have claimed that brGDGT distribution in lake sediments differed from catchment soils as we do in Gonghai Lake in this study.

As to others lakes, due to the lack of catchment soil brGDGT data, they are not shown in revised Figure 5, although brGDGT-derived temperatures are also warm season biased. We have added related content in the revised manuscript. Please see line 358-381 in the revision.

**1319-325 Rephrase**

Response: Done. Please see line 371-381 in the revision.

1332 You should also mention that brGDGT distribution in water column seems to change seasonally in agreement with temperature, what you discussed in the 4.3 section.

Response: Although the MBT'5ME and MBT'6ME in water SPM mirror the water temperature change in Sept and Jan, the calculated seasonal temperature offsets was quite small in Gonghai Lake. So we don't emphasize this phenomenon. We have added related content in the revised manuscript. Please see line 391-409 in the revision.

**1339 Rephrase**

Response: Done. Please see line 454-456 in the revision.

1350 Most of stratified lakes . . .

Response: Done. This paragraph has been rephrased and this word has been deleted.

1365 Can you add a reference?

Response: Done. Please see line 486 in the revision.

1378 universal

Response: Changed. Please see line 499 in the revision. 1383 compatible with the mechanism that we propose here Response: Changed. Please see line 504 in the revision. 1391 Use cold season rather than 'winter'. Response: This paragraph has been deleted. Conclusion 1403 from that in soils Response: Changed. Please see line 528 in the revision. 1403-404 indicating that lacustrine brGDGTs are mainly produced in situ . . . Response: Changed. Please see line 528 in the revision. 1404 in surface sediments of Lake Gonghai Response: Changed. Please see line 534 in the revision. 1406 water-air decoupling in Lake Gonghai Response: We don't add "in Lake Gonghai" here because we think the sentence is ok. Figure 1 (a) northern limit? (b) For the latitude replace 'E' by N 1655 represent Response: Changed. Please see in the revised Figure 1. Figure 3 fractional abundance Maybe specify: water column (SPM). Response: Changed. Please see line813 in the revision. 1667 Use degree of cyclisation rather than 'cyclisation ratio'. Response: Done. Please see line 813-814 in the revision. Figure 4 1670 soil calibration. 13

Response: This figure have been replace, please see revised caption Figure 4.

Figure 5

1679 A comma missing before Lake Donghu.

Response: Changed. Please see line 830 in the revision.

1676 Mention the number of the equation used.

Response: Done. Please see line 825-831 in the revision.

Table 1

1695 For b et c, mention the number of the equation used.

Response: Done. Please see the note of Table 1 in the revision.

Table 3

1705-708 Mention the number of the equation used.

Response: Done. Please see the note of Table 3 in the revision.

**Anonymous Referee #2**

Received and published: 2 March 2020

**General comments**

The mechanism of season bias of lake brGDGTs-derived temperature is not very clear, hence limit the application of brGDGTs index in lakes. The manuscript proposes a new idea about this hot topic. They conclude that decoupling of water and air temperature in winter causes warm season bias of lacustrine brGDGTs temperature estimates. Therefore, I recommend this manuscript for publication in the journal after improvement.

Response: Thanks for the comments. We have made substantial improvements according to reviewers' suggestions. Besides, some improvements were made beyond those suggestions during our revision. Please see the response and revised manuscript below.

**Detailed comments**

1/ Many pervious researchers suggested that soil calibrations could not be applicable to lake sediments for temperature reconstruction, if aquatic production of brGDGTs is predominant over soil input (e.g. many papers). It is no new, and not necessary to discuss too much in this point in your manuscript. And to focus on SPM.

Response: We agree. Some related sentences have been deleted and some related content has merged into the Section 4.1, as evidences of the in situ production of brGDGTs in the lakes. Please see Line 284-290 in the revision.

2/ Seasonality is a major feature for almost all organic proxies. For example, Lake Huguangyan (Hu et al., 2016; Chu et al., 2017). Lake limnology is most important, for example, Lake Huguangyan is a monomictic lake.

Response: We agree. Seasonality has been discussed in lines 421-444 in our manuscript and we are inclined to rule it out as a cause in our case. Lake Huguangyan has been used as a reference in many places in our paper. However, due to its location in the tropical area it is not the focus of our discussion. We give a special mention on proxy seasonality in the Lake Huguangyan. Moreover, we propose deep/bottom waters might influence brGDGT temperature signal in the lake. Please see lines 467-483 in the revision.

3/ "Line 147-148: "There is no water column stratifification whether summer or winter". You must revise this sentence. Based on the location and depth of the lake, it might be stratifified in summer. And figure 2 shows a little stratifification occurred in September (autumn).

Response: You are right. We changed these sentences. Please see lines 133-135 in the revision.

4/ Line 360-365: I don't think the estimated temperature using the calibration of Dang et al. (2018) are close to the mean warm season AT in GH, even if the RMSE is being considered. It seems that the calibration of Russell et al. (2018) may be more suitable for your explanation, and you'd get more discuss about this point.

Response: You are right. We rephrase these sentences. Please see lines 309-315 in the revision.

5/ Line 450: The definition of warm season should be given earlier, and change "monthly temperature" to "average monthly temperature".

Response: Done. The warm season is defined in the head of Section 4.2 '4.2 Lacustrine brGDGT-derived AT are warm season biased (average monthly temperature >0 °C)'. Please see line 300-301, 313 in the revision.

6/ Line 464-465: "For example, MBT/CBT-derived temperature correlated better with warm season AT than with annual mean AT in the tropical Lake Huguangyan, suggesting a warm season bias (Sun et al., 2011)". To improve the discussion of seasonality in the paper, I recommend authors should detailed read the paper of Sun et al. (2011) carefully. And the author should see discussion about the seasonality of brGDGTs in Lake Huguangyan from Hu et al. (2016) and Chu et al. (2017). Seasonal biases may be due to seasonal brGDGTs production, and link to lake limnology and local climate.

Response: Thanks for the comment. We misunderstood the results from the Lake Huguangyan and made changes accordingly. The sentence has been corrected as 'The MBT/CBT-derived temperature in the Lake Huguangyan was thought to reflect mean annual AT (Hu et al., 2015, 2016); however, has recently been proposed to be winter/cool biased (Chu et al., 2017)'. Lake Huguangyan is located in the tropical region, which is not the focus of our discussion. Nonetheless, we give a special mention

on proxy seasonality of Lake Huguangyan and other tropical lakes in second paragraph in Section 4.4. Please see line 470-475 in the revision.

7/ Please provide the component specific content of brGDGT as a Supplement.

Response: The brGDGT data had been showed in the data repository as journal recommends, please see https://figshare.com/s/a4f324247ecd9d1ac575.

8/ This manuscript is worth publish because something is new. But, authors should mention that the limited data in your manuscript, and more works are need to verify this question.

[revised manuscript text omitted]
                                                                                                                                                                                                                                                                                                                                                                                                                                                                                                                           |
| <li>276</li><li>277</li><li>278</li><li>279</li>                                                          |  <li>4.2 Soil brGDGTs reflect mean annual AT</li> <li>The in situ production of brGDGTs in the Gonghai Lake can be also evidenced by the</li> <li>discrepancies in reconstructed temperatures between soils and sediments/SPM. Based on the new</li> <li>global soil calibration of Eq. (9) and regional soil calibration of Eq. (10) for China, the</li>                                                                                                                                                                                                                                                                                                                                                                                 |
|  <li>276</li> <li>277</li> <li>278</li> <li>279</li> <li>280</li>                                         |  <li>4.2 Soil brGDGTs reflect mean annual AT</li> <li>The in situ production of brGDGTs in the Gonghai Lake can be also evidenced by the discrepancies in reconstructed temperatures between soils and sediments/SPM. Based on the new global soil calibration of Eq. (9) and regional soil calibration of Eq. (10) for China, the brGDGT-derived AT in the Gonghai catchment soils ranged from 1.18 to 2.75 °C (average 2.33 ±</li>                                                                                                                                                                                                                                                                                                      |
|  <li>276</li> <li>277</li> <li>278</li> <li>279</li> <li>280</li> <li>281</li>                            |  <li>4.2 Soil brGDGTs reflect mean annual AT</li> <li>The in situ production of brGDGTs in the Gonghai Lake can be also evidenced by the discrepancies in reconstructed temperatures between soils and sediments/SPM. Based on the new global soil calibration of Eq. (9) and regional soil calibration of Eq. (10) for China, the brGDGT-derived AT in the Gonghai catchment soils ranged from 1.18 to 2.75 °C (average 2.33 ± 0.65 °C; Table 1Fig. 4a) and from -4.22 to -1.21 °C (average -2.42 ± 1.19 °C; Table 1), respectively.</li>                                                                                                                                                                                                |
|  <li>276</li> <li>277</li> <li>278</li> <li>279</li> <li>280</li> <li>281</li> <li>282</li>               |  <li>4.2 Soil brGDGTs reflect mean annual AT</li> <li>The in situ production of brGDGTs in the Gonghai Lake can be also evidenced by the discrepancies in reconstructed temperatures between soils and sediments/SPM. Based on the new global soil calibration of Eq. (9) and regional soil calibration of Eq. (10) for China, the brGDGT-derived AT in the Gonghai catchment soils ranged from 1.18 to 2.75 °C (average 2.33 ± 0.65 °C; Table 1Fig. 4a) and from -4.22 to -1.21 °C (average -2.42 ± 1.19 °C; Table 1), respectively. Considering the ±4.8 °C uncertainty of the global calibration and ±2.5 °C of the regional calibration,</li>                                                                                         |
|  <li>276</li> <li>277</li> <li>278</li> <li>279</li> <li>280</li> <li>281</li> <li>282</li> <li>283</li>  |  <li>4.2 Soil brGDGTs reflect mean annual AT The in situ production of brGDGTs in the Gonghai Lake can be also evidenced by the discrepancies in reconstructed temperatures between soils and sediments/SPM. Based on the new global soil calibration of Eq. (9) and regional soil calibration of Eq. (10) for China, the brGDGT-derived AT in the Gonghai catchment soils ranged from 1.18 to 2.75 °C (average 2.33 ± 0.65 °C; Table 1Fig. 4a) and from -4.22 to -1.21 °C (average -2.42 ± 1.19 °C; Table 1), respectively. Considering the ±4.8 °C uncertainty of the global calibration and ±2.5 °C of the regional calibration, the estimated temperatures from the global calibration are much close to the mean annual AT of </li>  |

| 285 | calibration Eq. (9) was applied to sediment/SPM data, yielding estimated temperatures $-0.50 \pm$                  |
|-----|--------------------------------------------------------------------------------------------------------------------|
| 286 | 0.78 °C in surface sediments and -0.55 <math>\pm</math> 0.52 °C in SPM and hence much lower than those from |
| 287 | surface soils (2.33 $\pm$ 0.65 °C; Table 1). Similarly, temperature underestimation using soil-derived             |
| 288 | calibration has been widely reported in many modern lake sediments (e.g., Tierney et al., 2010;                    |
| 289 | Loomis et al., 2012; Pearson et al., 2011; Russell et al., 2018), which has been attributed to in situ             |
| 290 | production of brGDGTs in the lakes.                                                                                |
| 291 | For some lakes, soil brGDGTs input may be significant and predominant over aquatic production,                     |
| 292 | yielding similar brGDGTs composition distributions between lake sediments and surrounding soils. In-               |
| 293 | such cases, soil calibrations could be still applicable to lake sediments for AT reconstruction-                   |
| 294 | (Niemann et al., 2012; Li et al., 2017; Ning et al., 2019; Tian et al., 2019). In our results using-               |
| 295 | soil-derived calibration of Eq. (9), the estimated temperatures from surface sediments ( $-0.50 \pm$               |
| 296 | 0.78  °C; Fig. 4a) and SPM (-0.55 ± 0.52 °C; Fig. 4a) were much lower than those from surface soils-               |
| 297 | (2.33 ± 0.65 °C; Fig. 4a). Similarly, temperature underestimation has been widely reported in many-                |
| 298 | modern lake sedimentsglobal lakes (e.g., Tierney et al., 2010; Loomis et al., 2012; Pearson et al., 2011;          |
| 299 | Russell et al., 2018), which is likely associated with in situ production of brGDGTs in the lakes.                 |
| 300 | 4.2 Lacustrine brGDGT -derived AT s are warm season biased (average monthly                          |
| 301 | temperature >0 °C)                                                                                       |

302 The above evidence suggests that the application of temperature calibrations based on soil-

| 303 | brGDGTs (by De Jonge et al. (, 2014)) to lake sediments is risky. Therefore, The suggested in situ                        |
|-----|---------------------------------------------------------------------------------------------------------------------------|
| 304 | production of brGDGTs prompts us to use lake-specific temperature calibrations (Tierney et al., 2010;                     |
| 305 | Pearson et al., 2011; Sun et al., 2011; Loomis et al., 2012; Dang et al., 2018; Russell et al., 2018) to                  |
| 306 | reconstruct AT, although not differentiated quantitatively the relative contributions of aquatic vs.                      |
| 307 | soil-derived brGDGTs. Here, we applied four equations, Eqs. (11) and (15)-(17) in Table 2, to our                         |
| 308 | sedimentary brGDGT data.                                                                                                  |
| 309 | As shown in Fig. 4a, the reconstructed temperatures using different equations are >6.4 °C.                                |
| 310 | Despite discrepancies in the temperature values between calibrations, they are comparable                                 |
| 311 | considering the uncertainty of each calibration. A prominent feature of the reconstructed temperature                     |
| 312 | is that they, especially those in the shallower sediments, are well above the annual mean AT but more                     |
| 313 | close to the mean warm season AT (average monthly temperature >0 °C). This feature is consistent                          |
| 314 | with numerous studies proposing that lacustrine brGDGT-derived ATs are warm season biased                                 |
| 315 | (Shanahan et al., 2013; Peterse et al., 2014; Dang et al., 2018).                                                         |
| 316 | In September, the values of MBT'5ME and MBT'6ME in SPM gradually decreased with depth,                                    |
| 317 | similar to the measured water temperature profile in the water column (Fig. 2). In January, the values                    |
| 318 | of MBT' SME and MBT' 6ME in SPM remained constant at different depths, also similar to the measured |
| 319 | water temperature profile in water column (Fig. 2). In addition, the values of MBT'5ME and MBT'6ME                        |
| 320 | in SPM in September were higher than in January, corresponding to the warmer water temperature in                         |

| 321 | September (Table 1Fig. 2). This suggests that brGDGTs in SPM can record lake water temperature-                           |
|-----|---------------------------------------------------------------------------------------------------------------------------|
| 322 | changes, as previously reported (Loomis et al., 2014; Hu et al., 2016; Zhang et al., 2016; Qian et al.,                   |
| 323 | 2019). Our results suggest both MBT'5ME and MBT'6ME could work wellrespond to indicate                                    |
| 324 | temperature changes to some extent. However, air temperature has been found to be correlated well-                        |
| 325 | with MBT' 5ME 5-methyl brGDGTs in African lakes (Russell et al., 2018), but with MBT' 6ME 6-methyl- |
| 326 | brGDGTs in East Asian lakes (Dang et al., 2018; Qian et al., 2019), which remains elusive.                                |
| 327 | Although the MBT' 5ME and MBT' 6ME in SPM in the lake seem to reflect temperature changes in-       |
| 328 | the water column to some extent, the differences of brGDGT-derived temperatures based on-                                 |
| 329 | lake-specific calibrations between September and January (~0.3 °C) the measured difference                                |
| 330 | (~13 °C). In fact, similar results have been also reported in other lakes. For example, in the Lower                      |
| 331 | King Pond, the calculated seasonal temperature difference in surface water SPM was 5.4 °C,                                |
| 332 | significantly smaller than the measured difference about 28.3 °C (Loomis et al., 2014); in the                            |
| 333 | Huguangyan maar lake, the brGDGT calculated seasonal temperature difference was 8 °C, also                                |
| 334 | significantly smaller than the measured difference about 16 °C (Hu et al., 2016). a long residence                        |
| 335 | time of SPM, although not exactly known, in the water column. which may imprint multi-seasonal                            |
| 336 | brGDGTs signals on the SPM, as previously reported in Lower King pond (Loomis et al., 2014). Such-                        |
| 337 | a scenario may lead to more "fossil" brGDGTs in SPM than those produced within a specific season-                         |
| 338 | or month, as evidenced by an observation showing that only a small proportion of intact polar lipid of                    |
| 339                                                                                                                | brGDGTs, indicative of fresh brGDGTs, was detected in total brGDGTs in SPM in a shallow lake-                                                                                                                                                                                                                                                                                                                                                                                                                                                                                                                                                                                                                                                                               |
|--------------------------------------------------------------------------------------------------------------------|-----------------------------------------------------------------------------------------------------------------------------------------------------------------------------------------------------------------------------------------------------------------------------------------------------------------------------------------------------------------------------------------------------------------------------------------------------------------------------------------------------------------------------------------------------------------------------------------------------------------------------------------------------------------------------------------------------------------------------------------------------------------------------|
| 340                                                                                                                | (Qian et al., 2019). Sediment resuspension, which may admix to the SPM that are both in-situ-                                                                                                                                                                                                                                                                                                                                                                                                                                                                                                                                                                                                                                                                               |
| 341                                                                                                                | produced and deposited from the water column, could be also important for smoothing the                                                                                                                                                                                                                                                                                                                                                                                                                                                                                                                                                                                                                                                                                     |
| 342                                                                                                                | temperature signal in SPM due to its shallow water depth (<10 m) and hence prone to be dynamic, as-                                                                                                                                                                                                                                                                                                                                                                                                                                                                                                                                                                                                                                                                         |
| 343                                                                                                                | evidenced by the lack of water column temperature stratification in the whole year (Fig. 2). Both-                                                                                                                                                                                                                                                                                                                                                                                                                                                                                                                                                                                                                                                                          |
| 344                                                                                                                | residence of "fossil" brGDGTs and sediment resuspension in SPM may cause the reduced seasonal                                                                                                                                                                                                                                                                                                                                                                                                                                                                                                                                                                                                                                                                               |
| 345                                                                                                                | difference in the estimated temperatures in SPM of Gonghai Lake. Besides, the indices such as IIIa/IIa,                                                                                                                                                                                                                                                                                                                                                                                                                                                                                                                                                                                                                                                                     |
| 346                                                                                                                | IR 6ME , #Rings tetra and #Rings penta in SPM were all in-between the soil and sediment values, suggestive-                                                                                                                                                                                                                                                                                                                                                                                                                                                                                                                                                                                                                                |
| 347                                                                                                                | of more impact of soil input on brGDGTs in SPM than in sediments, which could also reduce the                                                                                                                                                                                                                                                                                                                                                                                                                                                                                                                                                                                                                                                                               |
|                                                                                                                    |                                                                                                                                                                                                                                                                                                                                                                                                                                                                                                                                                                                                                                                                                                                                                                             |
| 348                                                                                                                | seasonal contrast in estimated temperatures.                                                                                                                                                                                                                                                                                                                                                                                                                                                                                                                                                                                                                                                                                                                                |
| 348
349                                                                                                         | seasonal contrast in estimated temperatures.
Many previous brGDGT instrumental analyses on lake materials used one cyano column, which                                                                                                                                                                                                                                                                                                                                                                                                                                                                                                                                                                                                                                   |
| <li>348</li><li>349</li><li>350</li>                                                                      | seasonal contrast in estimated temperatures.
Many previous brGDGT instrumental analyses on lake materials used one cyano column, which
did not separate 5- and 6-methyl brGDGTs. Using the data published in the same lake from Cao et al.                                                                                                                                                                                                                                                                                                                                                                                                                                                                                                                            |
| <li>348</li><li>349</li><li>350</li><li>351</li>                                                          |  <li>seasonal contrast in estimated temperatures.</li> <li>Many previous brGDGT instrumental analyses on lake materials used one cyano column, which did not separate 5- and 6-methyl brGDGTs. Using the data published in the same lake from Cao et al.</li> <li>(2017), we re-calculated temperature using different calibrations. The results showed that the absolute</li>                                                                                                                                                                                                                                                                                                                                                                                     |
|  <li>348</li> <li>349</li> <li>350</li> <li>351</li> <li>352</li>                                         | seasonal contrast in estimated temperatures.
Many previous brGDGT instrumental analyses on lake materials used one cyano column, which
did not separate 5- and 6-methyl brGDGTs. Using the data published in the same lake from Cao et al.
(2017), we re-calculated temperature using different calibrations. The results showed that the absolute
temperature estimates were all significantly warmer than the mean annual AT (Table 3), with the                                                                                                                                                                                                                                                                                                              |
|  <li>348</li> <li>349</li> <li>350</li> <li>351</li> <li>352</li> <li>353</li>                            |  <li>seasonal contrast in estimated temperatures.</li> <li>Many previous brGDGT instrumental analyses on lake materials used one cyano column, which did not separate 5- and 6-methyl brGDGTs. Using the data published in the same lake from Cao et al. (2017), we re-calculated temperature using different calibrations. The results showed that the absolute temperature estimates were all significantly warmer than the mean annual AT (Table 3), with the temperature offsets varying from 4–10 °C, which cannot be fully explained by the uncertainty of each</li>                                                                                                                                                                                         |
|  <li>348</li> <li>349</li> <li>350</li> <li>351</li> <li>352</li> <li>353</li> <li>354</li>               | seasonal contrast in estimated temperatures.
Many previous brGDGT instrumental analyses on lake materials used one cyano column, which
did not separate 5- and 6-methyl brGDGTs. Using the data published in the same lake from Cao et al.
(2017), we re-calculated temperature using different calibrations. The results showed that the absolute
temperature estimates were all significantly warmer than the mean annual AT (Table 3), with the
temperature offsets varying from 4–10 °C, which cannot be fully explained by the uncertainty of each
calibration. Therefore, it appears that sedimentary brGDGT-derived temperature is warm season                                                                                                     |
|  <li>348</li> <li>349</li> <li>350</li> <li>351</li> <li>352</li> <li>353</li> <li>354</li> <li>355</li>  | seasonal contrast in estimated temperatures.
Many previous brGDGT instrumental analyses on lake materials used one cyano column, which
did not separate 5- and 6-methyl brGDGTs. Using the data published in the same lake from Cao et al.
(2017), we re-calculated temperature using different calibrations. The results showed that the absolute
temperature estimates were all significantly warmer than the mean annual AT (Table 3), with the
temperature offsets varying from 4–10 °C, which cannot be fully explained by the uncertainty of each
calibration. Therefore, it appears that sedimentary brGDGT-derived temperature is warm season
biased in the Gonghai Lake irrespective of whether or not 5- and 6-methyl brGDGTs are separated. |

| 357 | sourced from aquatic production, Moreover, we found the warm season bias of reconstructed AT is          |
|-----|----------------------------------------------------------------------------------------------------------|
| 358 | increasingly apparent with the increase of latitude. Here, five lakes, including Lower King pond         |
| 359 | (Loomis et al., 2014), Qinghai Lake (Wang et al., 2012), Lake Donghu (Qian et al., 2019),                |
| 360 | Huguangyan maar (Hu et al., 2015, 2016) and Lake Towuli (Tierney and Russell, 2009), were selected       |
| 361 | to compare as an example. These lakes are located in different regions spanning a relatively large       |
| 362 | environmental gradient, and more importantly, brGDGT data from both the lake surface sediments           |
| 363 | and the surrounding soils are available. We re-calculated temperatures from published data of            |
| 364 | brGDGTs from these lakes (Fig. 5) by applying the calibration of global soils (Eq (8); Peterse et al.,   |
| 365 | 2012) to the surrounding soils and the calibration of lake surface sediments (Eq (11); Sun et al., 2011) |
| 366 | to the lake sediments. As shown in Fig. 5a, the brGDGT-inferred temperatures in catchment soils are      |
| 367 | similar to local mean annual ATs. In contrast, the brGDGT-inferred temperatures in lake sediments are    |
| 368 | similar to the local mean annual ATs only in low-latitude lakes, whereas they become increasingly        |
| 369 | higher than the local mean annual ATs toward higher latitudes (Fig. 5b). In comparison, the              |
| 370 | brGDGT-inferred temperatures are close to the local mean ATs in warm season (average monthly             |
| 371 | mean AT >0°C) in all these lakes (Fig. 5c). Besides above discussed lakes, Applying the global lake      |
| 372 | surface sediment calibration (Eq (10); Sun et al., 2011) to these lakes, we also re-calculated           |
| 373 | temperatures from published data of sedimentary brGDGTs (Fig. 5). Interestingly, the                     |
| 374 | brGDGTs-inferred temperatures were generally higher than the measured mean annual AT, with               |

| 375 | greater differences in higher latitude lakes (including the Gonghai Lake in this study) and close to the              |
|-----|-----------------------------------------------------------------------------------------------------------------------|
| 376 | mean annual AT in low-latitude or low-altitude lakes (i.e. the warm region; Fig. 5a). 
[revised manuscript text omitted]

| 501 | between AT and LWT caused by ice formation in winter may be applied to explain the observed                 |
|-----|-------------------------------------------------------------------------------------------------------------|
| 502 | seasonality of the brGDGT temperature records. For example, the biases of brGDGT-derived                    |
| 503 | temperatures toward summer AT observed extensively in the Arctic and Antarctic lakes (Shanahan et           |
| 504 | al., 2013; Foster et al., 2016) are compatible with the mechanism that we propose here. Of course,          |
| 505 | considering limited data in this study, more investigations are needed to test our viewpoint in future      |
| 506 | studies.                                                                                                    |
| 507 | We also noticed that the seasonality of brGDGTs-derived temperature occurs in some other lakes-      |
| 508 | also in tropical lakes.; however, there are disagreements in related studies. For example, sedimentary-     |
| 509 | MBT/CBTbrGDGT-derived temperature from lake-specific calibrations was unusual higher than Ning-             |
| 510 | et al., 2019).correlated better with warm season AT than with annual mean AT in the tropical Lake-          |
| 511 | Huguangyan, suggesting a warm season bias (Sun et al., 2011). However, the brGDGTs-inferred                 |
| 512 | temperatures reflect cold season temperature in some tropical lakes, such as Lake Challa, Lake Albert,      |
| 513 | Lake Edward and Lake Tanganyika (Tierney et al., 2010; Loomis et al., 2012; Buckles et al., 2014a).         |
| 514 | It is certain that the ice cover mechanism proposed here cannot be applied to thesethis tropical lakes-     |
| 515 | because ice cover does not form even in cold season winter except for high altitudes. In such cases, |
| 516 | other environmental conditions might determine the seasonality of brGDGT-based temperature-                 |
| 517 | proxies, such as seasonal soil erosion from lake catchments, seasonal production of brGDGTs and             |
| 518 | different production rate of brGDGTs at water depths (Sinninghe Damsté et al., 2009; Sun et al., 2011;      |

[revised manuscript text omitted]